# FOLLOW-YOUR-SHAPE: SHAPE-AWARE IMAGE EDITING VIA TRAJECTORY-GUIDED REGION CONTROL

**Zeqian Long** [2][†]**, Mingzhe Zheng** [1][†]**, Kunyu Feng**[†]**, Xinhua Zhang, Hongyu Liu** [1]**, Harry Yang** [1]**, Linfeng Zhang** [3]**, Qifeng Chen** [1]**, Yue Ma** [1][✉]

[1] HKUST, [2] University of Illinois Urbana-Champaign, [3] Shanghai Jiao Tong University

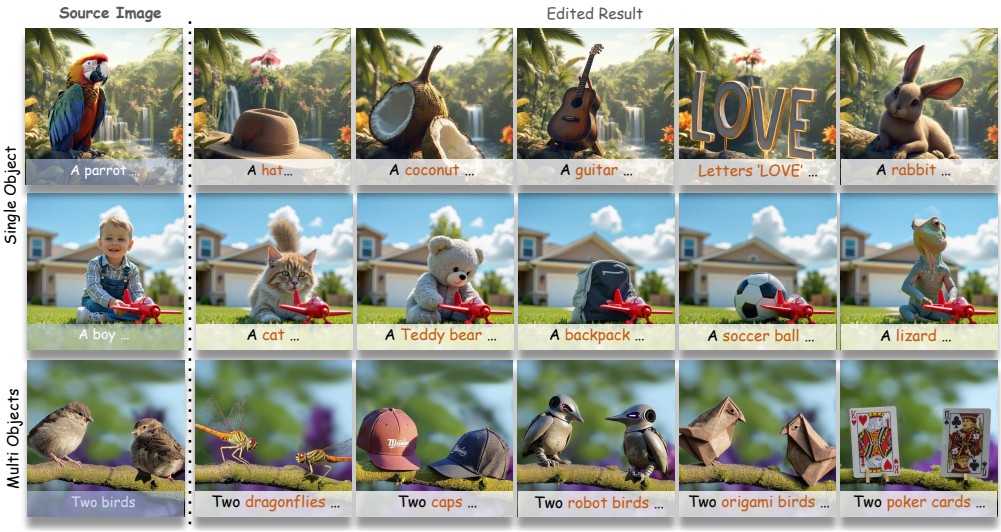

Figure 1: We propose **Follow-Your-Shape**, a training- and mask-free image editing framework that excels at prompt-driven shape transformation. Our method enables flexible modification of arbitrary object shapes while strictly maintaining non-target content. The examples demonstrate both single-object and multi-object cases involving significant shape transformation.

## ABSTRACT

While recent flow-based image editing models demonstrate general-purpose capabilities across diverse tasks, they often struggle to specialize in challenging scenarios—particularly those involving large-scale shape transformations. When performing such structural edits, these methods either fail to achieve the intended shape change or inadvertently alter non-target regions, resulting in degraded background quality. We propose **Follow-Your-Shape**, a training- and mask-free framework that supports precise and controllable editing of object shapes while strictly preserving non-target content. Motivated by the divergence between inversion and editing trajectories, we compute a **Trajectory Divergence Map (TDM)** by comparing token-wise velocity differences between the inversion and denoising paths. The TDM enables precise localization of editable regions and guides a **Scheduled KV Injection** mechanism that ensures stable and faithful editing. To facilitate a rigorous evaluation, we introduce *ReShapeBench*, a new benchmark comprising 120 new images and enriched prompt pairs specifically curated for shape-aware editing. Experiments demonstrate that our method achieves superior editability and visual fidelity, particularly in tasks requiring large-scale shape replacement.

---

† Equal contribution
✉ Corresponding author

# 1 INTRODUCTION

Recent advances in generative models have greatly expanded the scope of visual generation, enabling more controllable and realistic modifications across diverse scenarios. Image editing methods based on diffusion (Cao et al., 2023; Tumanyan et al., 2023; Feng et al., 2025) and flow models (Lipman et al., 2022; Labs, 2024; Labs et al., 2025; Kulikov et al., 2025) have demonstrated considerable success in general tasks, yet they often fail when faced with complex, large-scale shape transformations. These models can struggle to modify an object's structure as intended or may inadvertently alter background regions, which degrades the overall image quality. This limitation indicates a critical gap in their ability to perform precise structural edits while maintaining the integrity of unedited content.

The primary cause for this limitation lies in the inadequacy of existing region control strategies (Zhu et al., 2025; Cao et al., 2023). Methods that rely on external binary masks are often too rigid and struggle with the fine details of object boundaries. Alternatively, strategies that use cross-attention maps to infer editable regions are frequently unreliable, as these maps can be noisy and inconsistent. While unconditional Key-Value (KV) injection can preserve background structure, it lacks selectivity and tends to suppress the intended edits (Avrahami et al., 2025; Wang et al., 2024). We argue that a breakthrough requires a new approach: one that derives the editable region dynamically from the editing process itself by analyzing how the model's behavior shifts between the source and target conditions.

To address these challenges, we propose **Follow-Your-Shape**, a training- and mask-free framework for precise and controllable shape editing. As illustrated in Figure 2, the core innovation of our pipeline is the **Trajectory Divergence Map (TDM)**. The TDM is generated by computing the token-wise difference between the denoising velocity fields of the source and target prompts. This map accurately localizes the regions intended for editing, which in turn guides a selective KV injection mechanism to ensure that modifications are applied precisely where needed while preserving the background.

However, directly applying TDM-guided injection across all denoising timesteps is suboptimal because the TDM can be unstable in the early, high-noise stages of the process. We therefore introduce a **Scheduled KV Injection** strategy that adapts its guidance throughout the denoising process. As visualized in Figure 2, this staged approach first performs unconditional KV injection to stabilize the initial trajectory, and only then applies TDM-guided editing once a coherent latent structure has formed. This scheduling and staged editing pipeline ensures a more robust and faithful editing outcome compared to a direct application.

To validate our approach, we introduce *ReShapeBench*, a new benchmark with paired images and refined text prompts specifically designed for evaluating large-scale shape modifications. Beyond this new dataset, we further evaluate Follow-Your-Shape on the public *PIE-Bench* (Ju et al., 2023) to ensure generalizability. Follow-Your-Shape achieves state-of-the-art performance on both benchmarks, demonstrating superior background preservation, text–image alignment, and overall visual quality, confirming its effectiveness in both shape-aware and general editing tasks.

**Our primary contributions are summarized as follows:**

- A novel and training-free editing framework, **Follow-Your-Shape**, that utilizes a **Trajectory Divergence Map (TDM)** to achieve precise, large-scale shape transformations while preserving background content.

- A trajectory-guided **scheduled injection strategy** that improves editing stability by adapting the guidance mechanism throughout the denoising process.

- A new benchmark, *ReShapeBench*, designed for the systematic evaluation of shape-aware image editing methods.

**Reproducibility:** We provide our benchmark dataset and source code[1] to facilitate reproducibility and future research.

---

[1] https://github.com/mayuelala/FollowYourShape

## 2 RELATED WORK

**Region-Specific Image Editing.** A central challenge in image editing is localizing modifications to specific regions (Barnes et al., 2009; Zhang et al., 2024b; Liu et al., 2025a; Huang et al., 2025; Ma et al., 2025c;d; Long et al., 2025; Shen et al., 2025; Chen et al., 2025e; Feng et al., 2025). Early methods often relied on explicit user-provided masks to delineate editable areas (Lugmayr et al., 2022; Avrahami et al., 2023; Chen et al., 2023; Xiong et al., 2025; Wan et al., 2024). While effective for certain tasks, this approach requires manual annotation, limiting its applicability. To address this, subsequent work explored techniques to infer editable regions directly from text prompts. Methods such as Prompt-to-Prompt (Hertz et al., 2022) and Plug-and-Play (Tumanyan et al., 2023) manipulate cross-attention maps to associate textual tokens with spatial areas, enabling localized edits without explicit masks. Other approaches, such as DiffEdit (Couairon et al., 2022), generate a mask by computing differences between diffusion model predictions conditioned on source and target prompts. However, attention-based localization can be imprecise and unstable, especially during large-scale shape transformations where object boundaries change significantly (Pang et al., 2024; Cao et al., 2023). In contrast, Follow-Your-Shape provides a training-free and mask-free method for identifying editable regions directly from the model's behavior, avoiding the need for external masks or noisy attention maps.

**Structure Preservation via Inversion and Feature Reuse.** Preserving non-target regions is equally critical for high-fidelity editing, and this is closely tied to the quality of the model's inversion process. For diffusion models, significant research has focused on improving DDIM inversion (Song et al., 2020) to better reconstruct a source image from noise. Previous works like null-text inversion (Mokady et al., 2023) and optimization-based methods (Wallace et al., 2023) aim to reduce the discrepancy between the reconstruction and editing trajectories. With the shift toward flow-based models, inversion fidelity has become even more important due to their deterministic nature. RF-Inversion (Rout et al., 2024) formulates the inversion process as a dynamic optimal control problem, while RF-Solver (Wang et al., 2024) achieve more accurate reconstructions by incorporating higher-order derivative information. Beyond improving inversion, another line of work focuses on explicitly reusing modules or features from the source image's generation process (Zheng et al., 2024; Ma et al., 2025a; Yan et al., 2025). Techniques based on Key-Value (KV) caching (Zhu et al., 2025; Avrahami et al., 2025) or feature injection (Wang et al., 2024; Feng et al., 2025) enforce structural consistency by propagating source-image features into the new generation process. In contrast to prior methods that rely on simple heuristics, Follow-Your-Shape employs a trajectory-guided scheduled injection strategy to achieve more precise, content-aware control.

## 3 METHODOLOGY

Our goal is to enable precise object shape-aware editing while strictly preserving the background. Motivated by the limitations of existing region control strategies and the need for a more adaptive mechanism, we introduce Trajectory Divergence Map (TDM) that quantifies token-wise semantic deviation between inversion and editing trajectories, as shown in Figure 2. The overall pipeline of Follow-Your-Shape is shown in Figure 3.

### 3.1 MOTIVATION

Effective image editing requires a precise balance between introducing new content and preserving the original structure. As illustrated in Figure 2 (left), traditional structure-preserving editing approaches often produce unstable denoising trajectories that deviate significantly from the stable reconstruction path, leading to severe structural degradation and undesired artifacts. Moreover, prior methods for localizing edits have notable drawbacks:

- **Binary Segmentation Masks:** Rely on external tools (Kirillov et al., 2023; Ronneberger et al., 2015), introducing overhead and a dependency on mask quality. Their rigid boundaries hinder large-scale shape changes and often produce artifacts.
- **Cross-Attention Masks:** Inferred from model's cross attention during the diffusion process, these maps are often noisy and inconsistent, proving unreliable for localizing edits, especially during significant shape transformations.

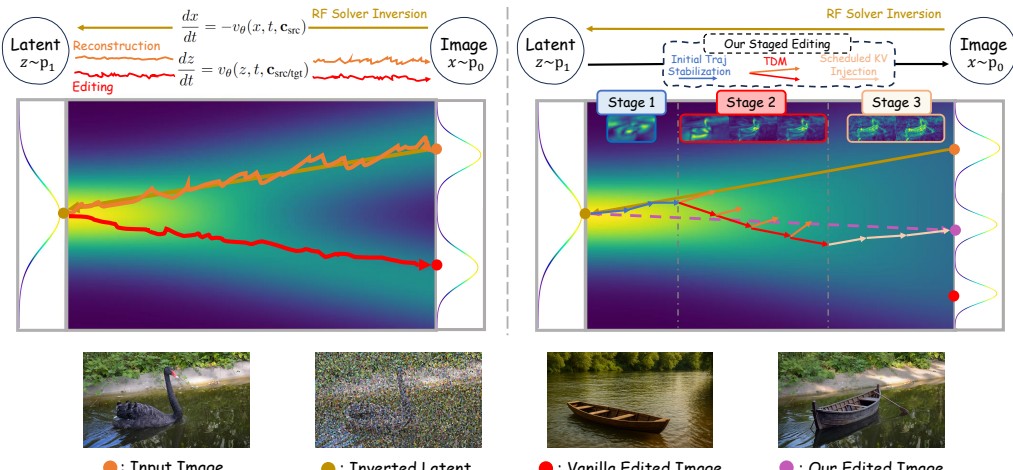

Figure 2: **Motivation for Trajectory Divergence Map (TDM) Guided Editing. Left:** Vanilla editing methods (red) often produce unstable trajectories compared to the stable reconstruction path (orange). **Right:** Our staged editing approach better resembles the ideal editing path. The TDM visualizes the dynamically localized editing region across different timesteps, with different border colors corresponding to different stages.

- **Unconditional Feature Injection:** This strategy preserves structure by globally injecting source features, but its lack of selectivity suppresses intentional edits, creating a conflict between editability and consistency.

To address these limitations, we propose a new approach from a dynamical systems perspective. We posit that the semantic difference between the source and target concepts can be measured by the divergence between their respective denoising trajectories. Based on this, we achieved a precise and mask-free method (shown in Figure 3) to stabilize the editing trajectory and perform targeted, shape-aware modifications without relying on external masks or rigid heuristics.

## 3.2 FOLLOW-YOUR-SHAPE

We perform shape-aware editing through a staged editing process that combines scheduled Key-Value (KV) injection with structural guidance, where the edit is localized by the Trajectory Divergence Map (TDM).

### 3.2.1 TRAJECTORY DIVERGENCE MAP

Our approach is grounded in the perspective of flow trajectories within the latent space, extending concepts from flow-matching frameworks to the inference setting. As illustrated in Figure 2 (left), a standard reconstruction follows a stable denoising trajectory guided by the source prompt $\mathbf{c}_{\text{src}}$. In an editing task, conditioning on a target prompt $\mathbf{c}_{\text{tgt}}$ alters the velocity field, causing the denoising trajectory to deviate from this initial path. We posit that the magnitude of this deviation spatially localizes the semantic difference between the two prompts. Regions intended for modification will exhibit significant divergence, while background areas will follow nearly identical trajectories. To formalize this, let $\{\mathbf{x}_t\}_{t=0}^T$ be the latent sequence from the source image inversion, and let $\{\mathbf{z}_t\}_{t=0}^T$ be the corresponding sequence during the editing (denoising) process. We define the token-wise **Trajectory Divergence Map (TDM)** $\delta_t$ at timestep $t$ as the $L_2$ norm of the difference between the velocity vectors predicted under the two prompts:

$$\delta_t^{(i)} = \left\| v_\theta(\mathbf{z}_t^{(i)}, t, \mathbf{c}_{\text{tgt}}) - v_\theta(\mathbf{x}_t^{(i)}, t, \mathbf{c}_{\text{src}}) \right\|_2, \tag{1}$$

where the velocity fields are evaluated at their respective trajectory latents, $\mathbf{z}_t$ and $\mathbf{x}_t$. To enhance interpretability and prepare the map for temporal aggregation, we apply min-max normalization

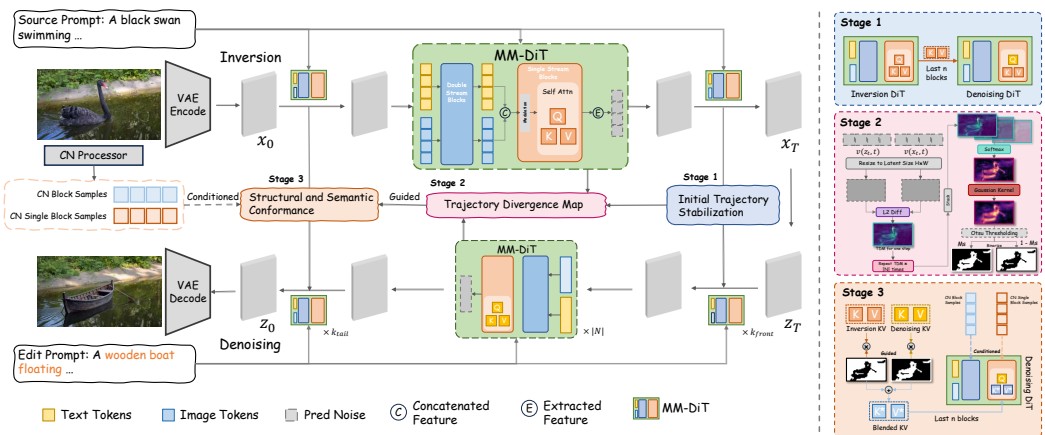

Figure 3: **Overview of our proposed pipeline.** Given a source image and the corresponding prompt, we first perform inversion to obtain the initial noisy latent code $x_T$. The editing process is then divided into three stages. *In Stage 1*, we stabilize the initial denoising trajectory by reusing key–value (KV) features recorded during inversion and injecting them into the early denoising steps. *In Stage 2*, we compute a Trajectory Divergence Map (TDM) by comparing denoising trajectories generated under the source and edit prompts, and process this map to precisely localize regions intended for editing. *In Stage 3*, guided by the TDM, blended KV features are injected into the final attention blocks to introduce new semantics while preserving non-target content. Simultaneously, ControlNet conditions are applied to provide auxiliary structural guidance.

across all spatial tokens $i$ at each timestep:

$$\tilde{\delta}_t^{(i)} = \frac{\delta_t^{(i)} - \min_j \delta_t^{(j)}}{\max_j \delta_t^{(j)} - \min_j \delta_t^{(j)}}. \tag{2}$$

As shown in Figure 2 (right), this produces a normalized TDM, $\{\tilde{\delta}_t^{(i)}\}$, which quantifies the localized editing strength on a scale of $[0, 1]$.

### 3.2.2 STAGED EDITING AND STRUCTURAL GUIDANCE

Directly applying TDM-guided injection across all timesteps is suboptimal due to the instability of the TDM in early, high-noise regimes (Figure 2 right). Early latents provide weak and noisy spatial signals, which can mislocalize edits if aggressive guidance is applied too soon. To address this, we introduce a scheduled injection strategy that partitions the $N$ denoising steps into three distinct phases and adapts the guidance mechanism to the latent state: the first phase emphasizes stabilization, the second collects and aggregates TDM evidence while allowing exploration, and the third enforces structural and semantic conformance.

**Stage 1: Initial Trajectory Stabilization.** For an initial set of $k_{\text{front}}$ timesteps, we perform unconditional KV injection from the source inversion path across all spatial tokens. This operation enforces a global reconstruction objective, equivalent to setting the edit mask $M_S = \mathbf{0}$, which stabilizes the trajectory and prevents semantic drift while the latent representation $\mathbf{z}_t$ is still dominated by noise. Intuitively, the model first anchors to a faithful reconstruction manifold before any region-specific modification is attempted, reducing the risk of spurious changes to background layout or texture.

**Stage 2: Editing and TDM Aggregation.** Once a stable latent structure has emerged, we begin the editing phase over a predefined window of timesteps $N$. During this window, we perform editing by setting the edit mask $M_S = \mathbf{1}$ at every step, allowing the model to explore target-guided generation path. Simultaneously, we compute and store the normalized TDMs $\tilde{\delta}_t$ at each timestep within $N$, capturing the trajectory divergence guided by the source and target prompts. After this editing window concludes, we aggregate the stored TDMs $\{\tilde{\delta}_t\}$ across time to construct a temporally consistent and spatially coherent edit mask. Specifically, throughout the denoising process, a token that appears unchanged at an individual timestep may still experience evolution at subsequent steps.

Therefore, to ensure that the aggregation faithfully captures such temporal dynamics, we employ a softmax-weighted temporal fusion for each token $i$:

$$\hat{\delta}^{(i)} = \sum_{t \in N} \alpha_t^{(i)} \cdot \tilde{\delta}_t^{(i)}, \quad \text{where} \quad \alpha_t^{(i)} = \frac{\exp(\tilde{\delta}_t^{(i)})}{\sum_{t' \in N} \exp(\tilde{\delta}_{t'}^{(i)})}. \tag{3}$$

To ensure spatial coherence and suppress noisy edges, the resulting map $\hat{\delta}$ is further refined via convolution with a Gaussian kernel $\mathcal{G}_\sigma$ to obtain $\tilde{M}_S \in [0,1]^{H \times W}$:

$$\tilde{M}_S = \mathcal{G}_\sigma * \hat{\delta}. \tag{4}$$

We observe that the distribution of values in $\tilde{M}_S$ typically exhibits a skewed unimodal shape (as shown in Figure 4), characterized by a dominant background mode and a long-tailed foreground response. Such a distribution is well suited for Otsu's method (Otsu et al., 1975), which selects the threshold $\tau$ that that maximizes the inter-class variance of values. Formally, for a candidate threshold $\tau$, let $\omega_0, \mu_0$ and $\omega_1, \mu_1$ denote the class probabilities and means of the background ($\tilde{M}_S \leq \tau$) and foreground ($\tilde{M}_S > \tau$), respectively. The between-class variance is defined as:

$$\sigma_b^2(\tau) = \omega_0(\tau)\,\omega_1(\tau)\left(\mu_0(\tau) - \mu_1(\tau)\right)^2. \tag{5}$$

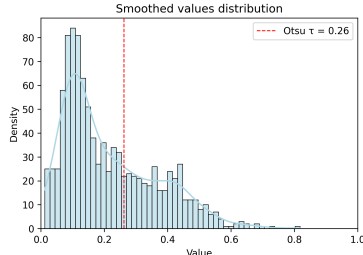

Figure 4: Distribution of values of $\tilde{M}_S$. The red dashed line indicates the Otsu threshold $\tau$.

The optimal threshold is then given by:

$$\tau^* = \arg\max_\tau \sigma_b^2(\tau). \tag{6}$$

The final binary mask $M_S$ is thus obtained by applying this threshold:

$$M_S = \mathbf{1}\left[\tilde{M}_S > \tau^*\right], \quad M_S \in \{0,1\}^{H \times W}, \tag{7}$$

where $\mathbf{1}[\cdot]$ denotes the indicator function.

**Stage3: Structural and Semantic Conformance.** Our framework enforces structural conformance by jointly leveraging TDM-guided feature injection for background preservation and ControlNet residual conditioning for stabilizing structural patterns. The mask $M_S$ obtained in Stage 2 modulates the fusion of Key-Value features, activating the target features ($K_{\text{tgt}}, V_{\text{tgt}}$) in edited regions and reverting to the source features ($K^{\text{inv}}, V^{\text{inv}}$) elsewhere. This feature-blending operation is formulated as:

$$\{K^*, V^*\} \leftarrow M_S \odot \{K^{\text{tgt}}, V^{\text{tgt}}\} + (1 - M_S) \odot \{K^{\text{inv}}, V^{\text{inv}}\}. \tag{8}$$

For structural guidance, ControlNet conditions the process on structural information $\mathbf{c}_{\text{cond}}$ by injecting a residual stream into each block of the denoising model $v_\theta$. For a latent representation $\mathbf{z}_t$ at a given block, the output $\mathbf{z}_t'$ is computed as:

$$\mathbf{z}_t' = \text{Block}(\mathbf{z}_t) + \beta \cdot \text{ControlNetBlock}(\mathbf{z}_t, \mathbf{c}_{\text{cond}}), \tag{9}$$

where $\beta$ controls the guidance strength. Concurrently, our feature injection mechanism builds on RF-Edit's background preservation by replacing the standard self-attention with a TDM-guided variant. The modified attention output $\boldsymbol{F}_{\text{out}}'$ is computed using the blended key-value pairs from Eq. 8:

$$\boldsymbol{F}_{\text{out}}' = \text{Attention}(Q^{\text{tgt}}, K^*, V^*). \tag{10}$$

This synergy between ControlNet's geometric enforcement and our TDM-guided semantic preservation enables precise, high-fidelity edits. The algorithmic implementation can be found in Algorithm 1 in Appendix.

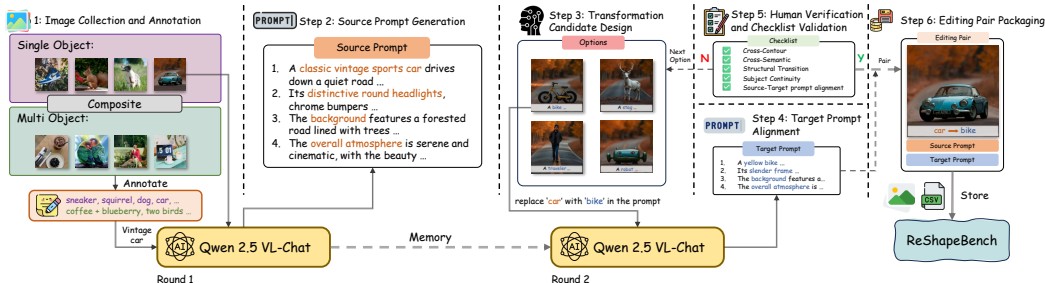

Figure 5: **Construction Process of *ReShapeBench*.** Note that images in Step 3 are generated after benchmark construction to serve as visual references. Checklist validation is performed on prompt.

## 4 RESHAPEBENCH: A BENCHMARK FOR LARGE-SCALE SHAPE TRANSFORMATIONS

**Overview.** Existing benchmarks for image editing (Wang et al., 2023; Ju et al., 2023; Zhang et al., 2023) are not tailored to the demands of shape-aware editing, where the goal is to change object geometry while preserving the surrounding background. In particular, *PIE-Bench* (Ju et al., 2023) (700 images) uses concise prompts that often lack spatial or structural detail, and it aggregates heterogeneous tasks (object replacement, stylization, background modification) rather than isolating shape transformation as a first-class target. These properties make it difficult to diagnose whether a method truly performs structural change or relies on side effects such as texture shifts or background re-synthesis. We therefore introduce *ReShapeBench*, a benchmark that centers on mask-free, prompt-driven shape transformation with paired prompts and controlled background settings. This design isolates the factors relevant to structural change and reduces confounds from style or background alterations, enabling a targeted and reproducible evaluation protocol. It also serves as a targeted complement to existing evaluation suites, providing shape-focused test cases that fill the gap left by current general-purpose editing benchmarks.

**Benchmark Construction.** *ReShapeBench* contains 120 newly collected images split into three subsets: 70 single-object scenes for precise shape editing, 50 multi-object scenes for targeted mask-free edits, and a general evaluation set of 50 images that combines samples from both subsets with curated *PIE-Bench* cases to assess generalization. All images are standardized to $512 \times 512$ to normalize spatial scale and reduce variability across methods and backbones. Each new image is paired with two distinct shape transformations, yielding 240 editing cases across the single- and multi-object subsets, plus 50 cases in the general set, for a total of 290 shape-aware editing cases; this pairing increases task coverage and controls difficulty by varying the magnitude of structural change. Source–target prompts follow a structured template and differ only in the foreground object description, which stabilizes text-to-image alignment while holding background fixed. All prompt pairs are generated by **Qwen-2.5-VL** (Bai et al., 2025) and validated by human raters to ensure alignment and that the transformation satisfies the predefined shape criteria; ambiguous cases are double-checked to maintain consistency. Figure 5 and the Appendix B present the construction procedure and sample cases, including the selection checklist and prompt schema used during curation.

## 5 EXPERIMENT

### 5.1 EXPERIMENTAL SETUP

We use the open-source FLUX.1-[dev] model (Labs, 2024) as the base and run all experiments in PyTorch on an NVIDIA A100 (40 GB). We set the number of denoising steps to 14, guidance scale to 2.0, and $k_{\text{front}}$ to 2. We evaluate both ControlNet-free and ControlNet-enabled variants of our method. If enabled, we apply multi-ControlNet conditioning with depth and Canny branches over the normalized denoising interval $[0.1, 0.3]$, with respective strengths 2.5 and 3.5. Unless otherwise stated, we keep the same inference scheduler and tokenizer as the official release and fix random

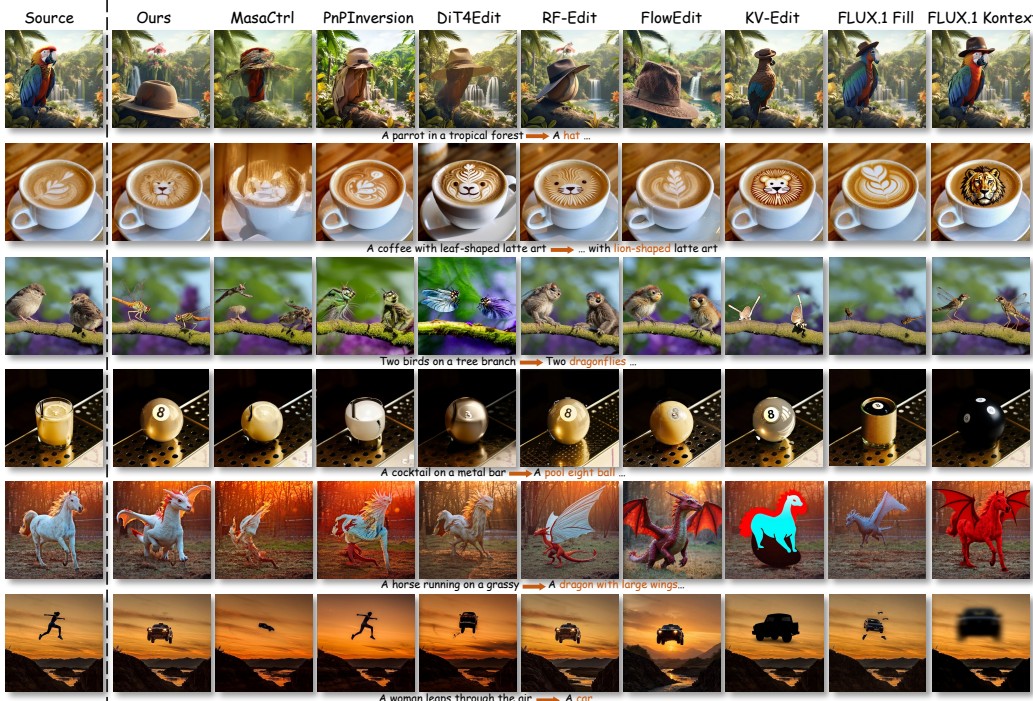

Figure 6: **Qualitative comparisons on various shape-aware editing cases.** Follow-Your-Shape successfully performs large-scale shape transformations while preserving the background, demonstrating advantages in both editing ability and visual consistency over existing baselines.

seeds for reproducibility; additional implementation details and runtime/memory profiles are provided in Appendix C.

## 5.2 COMPARISON WITH BASELINES

### 5.2.1 QUALITATIVE COMPARISON

We compare Follow-Your-Shape with *diffusion-based* and *flow-based* methods. Diffusion-based baselines include MasaCtrl (Cao et al., 2023), PnPInversion (Ju et al., 2023), and Dit4Edit (Feng et al., 2025), which modulate attention and conditions during the diffusion process. Flow-based baselines include RF-Edit (Wang et al., 2024), FlowEdit (Kulikov et al., 2025), KV-Edit (Zhu et al., 2025), FLUX.1 Fill (Labs, 2024), and FLUX.1 Kontext (Labs et al., 2025), which build on Rectified Flow. FLUX.1 Fill is designed for prompt-based masked image completion, while FLUX.1 Kontext leverages context-token concatenation for in-context editing. Figure 6 shows that Follow-Your-Shape achieves stronger shape-aware editing and background preservation. Diffusion-based methods tend to degrade the background under structural edits and may fail on high-magnitude shape changes, while flow-based methods produce higher-quality images but still exhibit detail jitter, ghosting, or incomplete transformations in difficult cases. Follow-Your-Shape performs large-scale shape transformations while preserving non-target regions.

### 5.2.2 QUANTITATIVE COMPARISON

We conduct quantitative evaluations on both *ReShapeBench* and *PIE-Bench* against diffusion- and flow-based baselines to assess both shape-aware editing and general editing performances. To ensure fairness, we use identical source and target prompts and the same number of denoising steps across methods. Because we follow RF-Solver with a second-order scheme, we double the number of steps for methods without a second-order update to match the number of function evaluations (NFE). We disable the ControlNet modules to isolate the effect of TDM-guided editing.

Table 1: **Quantitative comparison with state-of-the-art methods on *ReShapeBench* and *PIE-Bench*. Bold** and underlined values denote the best and second-best results respectively.

| Datasets | | *ReshapeBench* | | | | *PIE-Bench* | | |
|---|---|---|---|---|---|---|---|---|
| Metrics | Image Quality | Background Preservation | | Text Align | Image Quality | Background Preservation | | Text Align |
| Methods | AS ↑ | PSNR ↑ | LPIPS$_{\times 10^3}$ ↓ | CLIP Sim ↑ | AS ↑ | PSNR ↑ | LPIPS$_{\times 10^3}$ ↓ | CLIP Sim ↑ |
| MasaCtrl(Cao et al., 2023) | 5.83 | 23.54 | 125.36 | 20.84 | 5.61 | 21.58 | 130.71 | 19.53 |
| PnPInversion(Ju et al., 2023) | 6.11 | 24.77 | 102.91 | 19.23 | 5.94 | 22.69 | 108.43 | 24.62 |
| Dit4Edit(Feng et al., 2025) | 6.14 | 24.36 | 83.75 | 22.66 | 6.03 | 22.74 | 97.65 | 23.87 |
| RF-Edit(Wang et al., 2024) | 6.52 | 33.28 | 17.53 | 30.41 | 6.49 | 31.97 | 15.34 | 29.67 |
| FlowEdit(Kulikov et al., 2025) | 6.42 | 32.46 | 18.92 | 28.94 | 6.37 | 32.68 | 16.42 | 28.93 |
| KV-Edit(Zhu et al., 2025) | 6.51 | 34.73 | 16.42 | 26.97 | 6.47 | 33.45 | 13.72 | 28.14 |
| FLUX.1Fill(Labs, 2024) | 6.32 | 31.57 | 19.04 | 28.75 | 6.33 | 32.76 | 17.43 | 26.59 |
| FLUX.1Kontext(Labs et al., 2025) | 6.53 | 32.91 | 18.35 | 28.53 | 6.47 | 34.91 | 14.62 | 28.79 |
| Ours (w/o ControlNet) | 6.52 | 34.85 | 9.04 | 32.97 | 6.49 | 35.62 | 9.74 | 32.47 |
| **Ours** (Full Model) | **6.57** | **35.79** | **8.23** | **33.71** | **6.55** | **36.02** | **8.34** | **33.51** |

As shown in Table 1, we evaluate background consistency with PSNR (Huynh-Thu & Ghanbari, 2008) and LPIPS (Zhang et al., 2018), image quality with the LAION Aesthetic Score (Schuhmann et al., 2022), and text alignment with CLIP similarity (Radford et al., 2021). Appendix B.3 demonstrates the implementation details, including the preprocessing and metric computation settings used for all methods. Our method outperforms all baselines across metrics. Additionally, without the ControlNet module does not lead to a significant degradation in editing performance, indicating the improvements of our model are independent of this module. The region-controlled editing strategy improves fine-grained shape-aware editing while the mask $M_S$ preserves background content.

## 5.3 ABLATION STUDY

Initial trajectory stabilization and the timing and strength of ControlNet conditioning have the largest impact on editing performance. We therefore ablate these two components: the former regulates early trajectory stability, while the latter controls structural guidance during mid–late steps.

### 5.3.1 INITIAL TRAJECTORY STABILIZATION

To assess the role of initial trajectory stabilization, we vary the number of stabilization steps $k_{\text{front}}$ from 0 to 4. As shown in Figure 7 (i), small $k_{\text{front}}$ leads to drift and structural deviation, while large $k_{\text{front}}$ restricts the intended shape change. Table 2 shows that larger $k_{\text{front}}$ improves background preservation but reduces CLIP similarity, indicating a trade-off between stability and editability. $k_{\text{front}} = 2$ provides the best balance, yielding stable trajectories while maintaining sufficient freedom for large shape transitions.

Table 2: **Ablation study on different $k_{\text{front}}$.**

| $k_{\text{front}}$ | Image Quality | Background Preservation | | Text Align |
|---|---|---|---|---|
| | Aesthetic Score ↑ | PSNR ↑ | LPIPS$_{\times 10^3}$ ↓ | CLIP Sim ↑ |
| 0 | 6.51 | 32.79 | 10.04 | 31.05 |
| 1 | 6.55 | 34.38 | 9.88 | 32.56 |
| 2 | **6.57** | **35.79** | **8.23** | **33.71** |
| 3 | 6.52 | 31.25 | 10.52 | 29.41 |
| 4 | 6.48 | 30.41 | 12.37 | 27.66 |

### 5.3.2 CONTROLNET CONDITIONING TIMESTEP AND STRENGTH

To explore the effect of ControlNet conditioning timestep, we vary the injection interval within the normalized denoising range [0, 1]. Figure 7 (ii.a) shows that earlier injection yields better results, as latent features are less noisy and more receptive to structural guidance. We also vary the depth and Canny strengths. As shown in Figure 7 (ii.b), moderate values (e.g., (2.5, 3.5)) best balance structure preservation and editability, while overly weak or strong signals under- or over-constrain the edit. These results suggest that early, moderate guidance best stabilizes geometry without suppressing desired semantic changes in the edited regions.

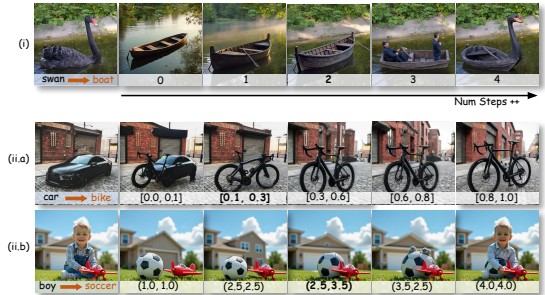
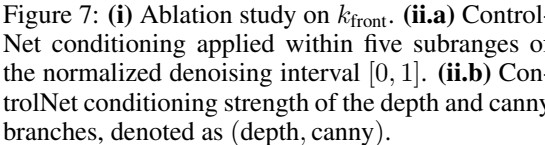
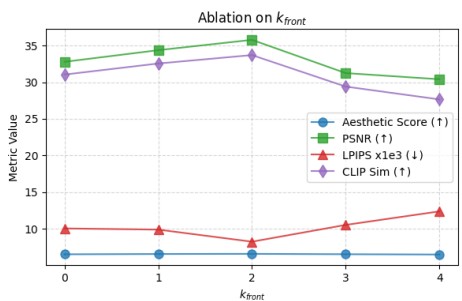

Figure 7: **(i)** Ablation study on $k_{\text{front}}$. **(ii.a)** Control-Net conditioning applied within five subranges of the normalized denoising interval $[0, 1]$. **(ii.b)** ControlNet conditioning strength of the depth and canny branches, denoted as (depth, canny).

Figure 8: **Visualization of ablation on** $k_{\text{front}}$. While the aesthetic score remains relatively stable, PSNR, LPIPS, and CLIP reveal a clear trade-off between editing strength and background preservation.

## 6 CONCLUSION

We introduce Follow-Your-Shape, a framework that enables large-scale object shape transformation by using a novel trajectory-based region control mechanism. Our method achieves precise, mask-free edits while preserving background integrity by dynamically localizing modifications through a Trajectory Divergence Map with scheduled injection. To properly evaluate this task, we developed *ReShapeBench*, a new benchmark tailored for complex shape-aware editing. To the best of our knowledge, Follow-Your-Shape is the first work to systematically address prompt-driven shape editing. Extensive qualitative and quantitative experiments validate its state-of-the-art performance on the proposed benchmark. Our work thus opens promising new avenues for controllable generation in image, video, and 3D content creation.

## ETHICS STATEMENT

Our method is designed for image editing tasks involving large shape transformations. To mitigate potential misuse such as malicious editing, the final model output incorporates an NSFW filtering component. The benchmark dataset used in this work is entirely collected from publicly available sources (https://www.pexels.com/), which explicitly permit free usage and modification of images and videos. A processed version of the dataset will be released solely for research purposes after further optimization. Beyond these considerations, our work does not involve any sensitive or personally identifiable data.

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

CONTENTS

# A PRELIMINARIES

## A.1 RECTIFIED FLOW (RF)

Let $p_0$ and $p_1$ denote the source and target distributions, respectively. Flow Matching (Lipman et al., 2022) models the transport between them by learning a time-dependent velocity field $v(t, x)$ that defines a continuous transformation $\psi_t(x)$ via the ordinary differential equation:

$$\frac{d\psi_t(x)}{dt} = v(t, \psi_t(x)), \quad \psi_0(x) \sim p_0, \ \psi_1(x) \sim p_1. \tag{11}$$

Rectified Flow (RF) (Liu et al., 2022) simplifies this by assuming a linear trajectory between $X_0 \sim p_0$ and $X_1 \sim p_1$:

$$X_t = (1 - t)X_0 + tX_1, \quad t \in [0, 1], \tag{12}$$

with the associated velocity field becomes:

$$v(X_t, t) = X_1 - X_0. \tag{13}$$

The model is trained by minimizing the conditional flow matching loss:

$$\mathcal{L}_{\text{CFM}} = \mathbb{E}_{X_0, X_1, t} \left[ \|v(X_t, t) - (X_1 - X_0)\|^2 \right]. \tag{14}$$

During inference, the learned velocity field is used to generate new samples by solving the reverse-time ODE:

$$\frac{dX_t}{dt} = -v(X_t, t), \tag{15}$$

starting from a sample $X_1 \sim \mathcal{N}(0, I)$. Since a closed-form solution is not available in general, we perform numerical integration over a discretized set of timesteps $\{t_i\}_{i=0}^N$. A standard choice uses first-order solvers such as Euler or Heun's method to approximate the trajectory:

$$X_{t_{i-1}} = X_{t_i} - h \cdot v(X_{t_i}, t_i), \tag{16}$$

where $h = t_i - t_{i-1}$ is the integration step size.

However, first-order solvers can suffer from numerical instability and truncation error, especially in high-dimensional generation tasks. Several recent works (Lu et al., 2022; Rout et al., 2024; Lv et al., 2025; Chen et al., 2025c; Wang et al., 2024) explore higher-order integration strategies or adaptive solvers to improve generation fidelity. Specifically, RF-Solver (Wang et al., 2024) introduces a second-order update derived from a Taylor expansion of the velocity field:

$$X_{t_{i-1}} = X_{t_i} - h \cdot v(X_{t_i}, t_i) + \frac{1}{2}h^2 \cdot \partial_t v(X_{t_i}, t_i), \tag{17}$$

where $\partial_t v(X_{t_i}, t_i)$ is the time derivative of the learned velocity field. This correction term reduces local integration error and leads to more accurate inversion and sampling trajectories, which is particularly important for downstream editing tasks that require high structural fidelity.

## A.2 KV INJECTION

Key-Value (KV) injection is adapted from the KV caching mechanism originally used in Transformers (Vaswani et al., 2017) to accelerate autoregressive inference (Pope et al., 2023). In large language models, cached key and value tensors allow reuse of past attention computations, enabling efficient decoding without recomputing earlier tokens.

When extended from language to vision models, KV reuse often generalizes beyond strict token caching. In U-Net based models (Cao et al., 2023; Qi et al., 2023), a common practice is to reuse intermediate attention maps or inject features derived from the inverted source image into self-attention layers. This feature-level injection plays a role similar to KV caching in LLMs by enforcing spatial consistency and anchoring the generative process to the source structure. In DiT-based architectures (Peebles & Xie, 2023), this idea extends to reusing value (V) matrices or full KV pairs, providing finer-grained control over how structural information is preserved during denoising.

To reduce memory cost and avoid limiting foreground flexibility, recent work such as Stable-Flow (Avrahami et al., 2025) explores the vital layers within DiT crucial for image formation. Therefore, by only reusing KV pairs in a subset of layers, it can balance structural fidelity and editability while effectively reducing memory usage.

In our work, we demonstrate that KV injection provides a modular and interpretable mechanism for controllable image editing and particularly effective in shape-aware tasks where edits must stay localized without affecting the broader scene.

## B   ADDITIONAL DETAILS ON RESHAPEBENCH CONSTRUCTION

### B.1   SHAPE TRANSFORMATION

While recent image editing models exhibit strong general-purpose editing capabilities, the concept of shape transformation remains ambiguous in the literature. In practice, object modifications usually include detail adjustments, color changes, or limited geometric variations, often relying on masks or ControlNet images; however, such operations cannot be explicitly framed as shape transformations. When constructing *ReShapeBench*, we need a clear definition and categorization of shape transformation to guide data curation and enable meaningful evaluation.

From a geometric perspective, shape transformation is a structural change beyond local affine operations such as scaling, rotation, or minor warping. It reconfigures the object's global contour and part topology, and may involve a shift in semantic class. At the same time, the transformed object must remain spatially coherent in the scene, occupying a similar anchor position and continuing to serve as the subject in context. Guided by these principles, we propose four criteria—cross-contour, cross-semantic, structural transition, and subject continuity—that together capture the essential properties of shape transformation.

- **Cross-contour:** The object's boundary undergoes a substantial change, exceeding local warping or affine resizing. This captures large-scale alterations to the external shape.
- **Cross-semantic:** The transformation shifts the object into a different semantic class, indicating a categorical rather than attributive change, while preserving overall scene coherence.
- **Structural transition:** The internal part topology is reconfigured, requiring modifications across multiple components instead of only simple attributes such as color or texture.
- **Subject continuity:** The transformed object retains its spatial anchor and role in the scene, remaining contextually consistent despite the change in shape and semantic.

Almost every editing case in the paper can be classified as shape transformation. Figure 9 provides an additional illustrative example. Note that we exclude posture or viewpoint changes (standing → sitting), as these involve articulation or perspective variation rather than structural transformation. The most challenging part in shape transformation is that it requires the model to localize and reinterpret object shape while maintaining consistency in background and composition. Unlike existing editing benchmarks that cover diverse editing tasks, our benchmark emphasizes large-scale shape transformation under prompt guidance, without relying on masks or external conditioning.

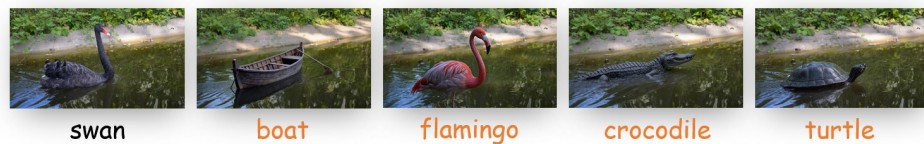

Figure 9: **Visualization of Shape Transformation**. The object's contour, semantic, structure changed while ensuring its subject continuity.

## B.2 IMAGE AND PROMPT EXAMPLES

As described in Section 3.2.2, the benchmark is constructed from collected images divided into single-object and multi-object categories. Specifically, single-object cases broadly cover four categories—nature, animals, indoor, and outdoor scenes; multi-object cases can also be categorized into indoor and outdoor scenes. Figures 10 and 11 provide representative samples. For each image, source and target prompts follow the four-sentence template, which is illustrated in Table 5, 6, and 7.

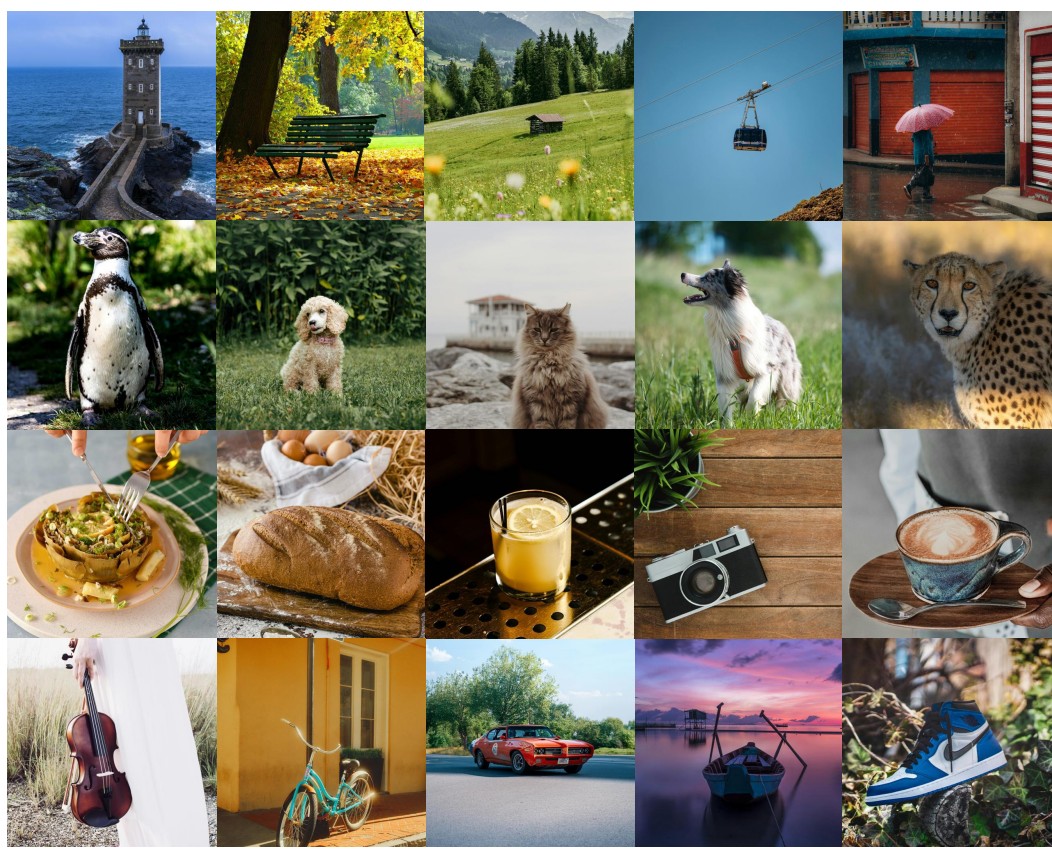

Figure 10: Single-Object Cases in *ReShapeBench*

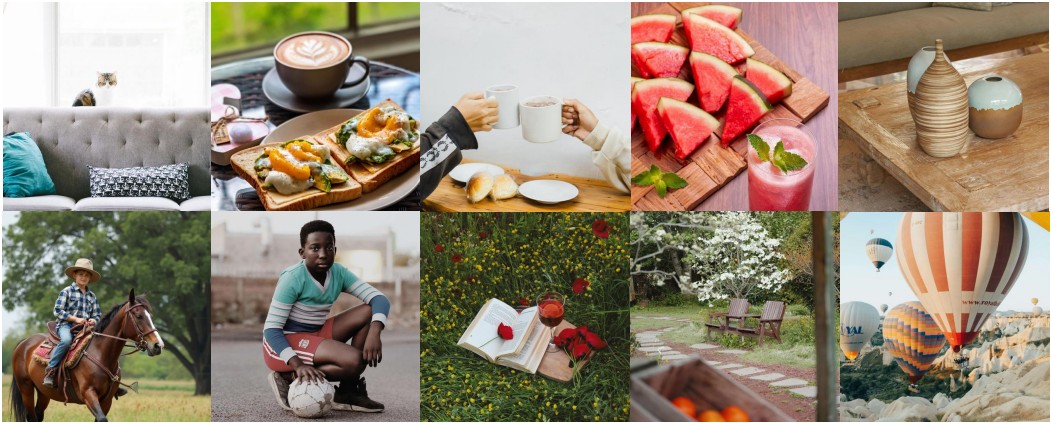

Figure 11: Multi-Object Cases in *ReShapeBench*

## B.3 EVALUATION METRICS

We use four metrics grouped under three aspects: image quality, background preservation, and text-image alignment.

Aesthetic Score (AS) measures the **perceptual quality** of the generated image by indicating how well the visual content conforms to natural image statistics. We compute AS with the LAION aesthetic predictor (Schuhmann et al., 2022), which applies a linear estimator on top of CLIP embeddings[2]. AS helps detect unnatural boundaries or blending artifacts that may occur during large-scale shape transitions and indicates how well the new shape integrates into the scene.

For **background preservation**, we adopt two widely used metrics that capture different aspects of visual similarity. Peak Signal-to-Noise Ratio (PSNR) measures low-level pixel fidelity, while Learned Perceptual Image Patch Similarity (LPIPS) evaluates perceptual similarity based on deep feature representations. Since we do not assume access to ground-truth masks and the edited shape can vary across models, we localize the edited region by applying a fixed-size box centered on the subject to occlude the foreground and compute similarity over the remaining background area. This heuristic enables fair evaluation of how well the unedited content is preserved.

Finally, to assess **text-image alignment**, we compute CLIP similarity between the generated image and the target prompt as an embedding-based measure of semantic consistency.

---

[2]https://github.com/LAION-AI/aesthetic-predictor

# C ADDITIONAL IMPLEMENTATION DETAILS

## C.1 PSEUDOCODE FOR FOLLOW-YOUR-SHAPE

---

**Algorithm 1** Region-Controlled Editing

---

**Input**: Inference steps $T$, Predicted velocities $\{v_{\text{src}}(\mathbf{x}_t^{(i)}), v_{\text{tgt}}(\mathbf{z}_t^{(i)})\}_{i=1}^T$, Source inversion features $\{K_t^{\text{inv}}, V_t^{\text{inv}}\}_{t=0}^T$, target prompt $\mathbf{c}_{\text{tgt}}$, schedule phase durations $\{k_{\text{front}}, k_{\text{tail}}\}$

1:   $N \leftarrow \{\}$                                                          ▷ Editing window set
2:   **for** $t = T$ down to 1 **do**
3:       **if** $t > T - k_{\text{front}}$ **then**
4:           $M_S \leftarrow \mathbf{0}$
5:       **else if** $T - k_{\text{front}} \geq t > k_{\text{tail}}$ **then**
6:           $\delta_t^{(i)} \leftarrow \left\| v_{\text{tgt}}(\mathbf{z}_t^{(i)}, t) - v_{\text{src}}(\mathbf{x}_t^{(i)}, t, \mathbf{c}_{\text{src}}) \right\|_2$
7:           $\tilde{\delta}_t^{(i)} \leftarrow \frac{\delta_t^{(i)} - \min_j \delta_t^{(j)}}{\max_j \delta_t^{(j)} - \min_j \delta_t^{(j)}}$                  ▷ TDM Computation
8:           $M_S \leftarrow \mathbf{1}$
9:           $N \leftarrow N \cup \{t\}$
10:     **else**
11:         $\hat{\delta}^{(i)} \leftarrow \sum_{t' \in T} \frac{\exp(\tilde{\delta}_{t'}^{(i)})}{\sum_{t'' \in T} \exp(\tilde{\delta}_{t''}^{(i)})} \cdot \tilde{\delta}_{t'}^{(i)}$
12:         $\tilde{M}_S \leftarrow \mathcal{G}_\sigma * \hat{\delta}$                                     ▷ TDM Aggregation
13:         $\tau \leftarrow \arg\max_{\tau'} \mathbb{P}[\tilde{M}_S \leq \tau'] \mathbb{P}[\tilde{M}_S > \tau'] \left( \mathbb{E}[\tilde{M}_S \mid \tilde{M}_S \leq \tau'] - \mathbb{E}[\tilde{M}_S \mid \tilde{M}_S > \tau'] \right)^2$
14:         $M_S \leftarrow \mathbf{1}[\tilde{M}_S > \tau]$
15:     **end if**
16:     $K^* \leftarrow M_S \odot K_t^{\text{tgt}} + (1 - M_S) \odot K_t^{\text{inv}}$
17:     $V^* \leftarrow M_S \odot V_t^{\text{tgt}} + (1 - M_S) \odot V_t^{\text{inv}}$
18:   **end for**
19:   **return** $K^*, V^*$

---

## C.2 HYPERPARAMETERS

All hyperparameters used in our experiments can be seen in Table 3. The table is divided into two groups: *General* and *Model Specific* hyperparameters. Since our method is training-free, the only general hyperparameters on FLUX are the inference step and the guidance scale. Note that because we adopt RF solver, and the solver skips the final timestep, the number of function evaluations (NFE) is $(15 - 1) \times 2 = 28$. We have already performed the ablation studies about $k_{\text{front}}$ and ControlNet parameters (Refer to Section 5.3). $k_{\text{tail}}$ controls the number of late timesteps where source features are injected; setting it to 2 or 3 ensures that the model has sufficient time to perform editing while still converging to the target distribution. The softmax scale regulates the sharpness of the aggregated TDMs, with a higher value producing a crisper mask, while Gaussian smoothing $\sigma$ removes spurious noise to ensure the continuity of the mask. The injecting DiT block indicates the starting index of feature injection, and we found that injecting from block 19 onward achieves the best trade-off between editing quality and memory efficiency. This hyperparameter is relatively flexible and can be slightly adjusted depending on the particular editing case. To ensure reproducibility, we also provide the exact hyperparameters used for the specific examples shown in the paper.

Table 3: **Hyperparameters**

| Hyperparameter | Value |
|---|---|
| *General* | |
| Inference Step | 15 |
| Guidance | 2 |
| *Model Specific* | |
| $k_{\text{front}}$ | 2 |
| $k_{\text{tail}}$ | 3 |
| Softmax scale (temperature) | 5 |
| Gaussian smoothing $\sigma$ | 0.7 |
| Injecting DiT block (start idx) | 19 |
| ControlNet Timing | [0.1, 0.3] |
| ControlNet Strength | (2.5, 3.5) |

Table 4: **Hyperparameters used for the results displayed in the paper.**

| Task | $k_{\text{front}}$ | $k_{\text{tail}}$ | CN type | CN timing | CN strength |
|---|---|---|---|---|---|
| parrot → hat, coconut, guitar, LOVE, rabbit | 2 | 3 | None | NA | NA |
| boy → cat, Teddy, backpack, soccer, lizard | 1 | 3 | Depth & Canny | [0.1, 0.3] | [2.5, 3.5] |
| bird → dragonflies, caps, robots, pokers | 2 | 3 | None | NA | NA |
| swan → boat, flamingo, crocodile, turtle | 3 | 3 | Depth | [0.1, 0.3] | 0.6 |
| leaf latte → lion, horse → dragon, cocktail → ball | 2 | 3 | None | NA | NA |

## C.3 RUNNING TIME AND MEMORY USAGE

**Running Time Analysis.** The method requires an additional diffusion pass each step to compute the second-order prediction, resulting in a total of 28 NFEs. As introduced in Section 5.1, we conduct our experiment on an NVIDIA A100 (40 GB), and the average running time for one image (averaged across multiple trials) is approximately 65.3 seconds. The computational cost is comparable to existing methods such as FlowEdit and RF-Solver-Edit. Since our work focuses on controllability and structural fidelity rather than acceleration, no additional optimization loops are introduced. Speed optimization is orthogonal to our contributions.

**Memory Usage.** During inference, the method stores a set of KV features and TDM maps on CPU memory, which amounts to approximately 12 GB in total. This cost is incurred only once during inversion and does not grow with the number of denoising steps. The GPU memory usage remains stable at around 25 GB throughout editing, comparable to existing flow-based editing pipelines. Since the method does not introduce any additional training procedures, this overhead is not a limiting factor in practice.

### C.4 POST-HOC ANALYSIS OF STRUCTURAL LOCALIZATION SIGNALS

To better understand whether the proposed Trajectory Divergence Map (TDM) provides a meaningful structural signal, we conduct a post-hoc analysis from two complementary perspectives: (1) mask-level comparison and (2) comparison against cross-attention maps.

**Mask-level comparison.** Our first goal is to verify that TDM indeed identifies the correct structural region to be edited. Using the crocodile case from Figure 9, we construct a pseudo ground-truth region by taking the union of the source mask and the target mask (generated by SAM (Kirillov et al., 2023)), and then downsampling it to the latent resolution. This union mask reflects the intuitive region of change: areas covered by either the original foreground or the edited foreground are exactly those that should be modified, while the background should remain unchanged. We compare this downsampled pseudo mask with our mask $M_S$. As shown in the top row of Fig 12, the two maps exhibit similarity in editable areas, providing an intuitive signal that TDM captures the correct semantic region for editing without relying on external supervision. This supports our claim that TDM provides an effective and interpretable estimate of "where" the model intends to apply shape transformation.

**TDM vs. cross-attention.** We further compare TDM with cross-attention maps in FLUX DiT. We visualize representative timesteps from the most responsive attention block. Because cross-attention activation is tightly tied to prompt tokens, its localization quality depends heavily on which word has the strongest response. In practice, identifying a single "correct" token is not feasible—responses vary significantly across tokens, heads, and layers—so we use a softmax-normalized variant over all text tokens. Even under this stabilized setting, cross-attention remains spatially noisy and useless with the true editing region. These attention maps are extracted when using TDM-guided region control. Maps without guided control are substantially noisier and therefore omitted for clarity. In contrast, TDM offers a much cleaner and more direct indication of structural change, and it is obtained in a far simpler manner by relying solely on trajectory differences.

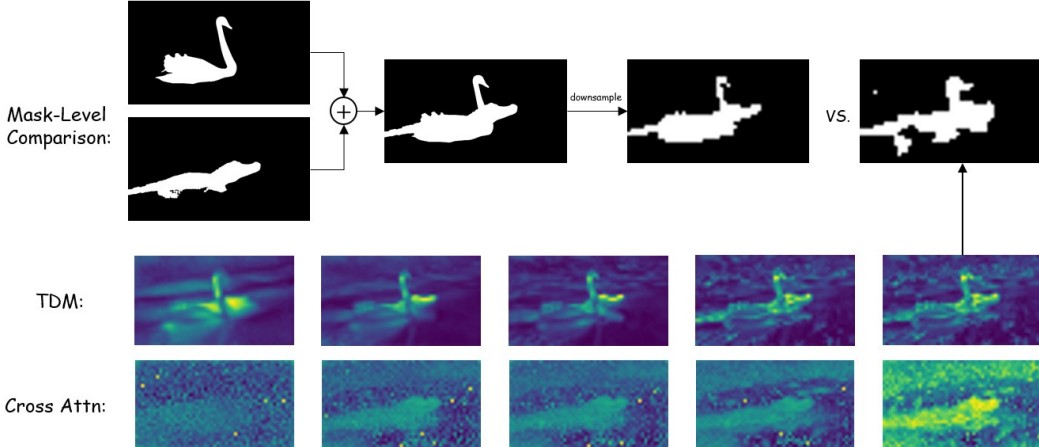

Figure 12: **Post-hoc analysis of structural localization signals.**

## D    LIMITATIONS AND FUTURE WORK

While our method demonstrates strong performance in shape-aware image editing, it also comes with certain limitations that suggest directions for future work.

### D.1    FAILURE CASES

Our method can be sensitive to prompt ambiguity and imprecise editing instructions. Since the editing behavior is driven entirely by prompt-guided inversion and denoising trajectories, the quality of the editing outcome depends on how clearly the intended modification is specified in the text prompt. When editing instructions are vague, lack clear semantic targets, or have low discriminative specificity, the model may struggle to determine where and how strongly to apply the modification. This often leads to weak, diffuse, or inconsistent edits, particularly in cases where the intended change is not explicitly specified by the prompt. For example, prompts that describe abstract transformations or rely on implicit assumptions about the editing target may result in edits that do not match user expectations (see Figure 13). Clear and well-defined editing descriptions that explicitly identify the object to be modified and the desired transformation are therefore important for reliable performance.

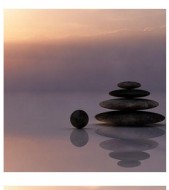

Source Prompt: A small stack of smooth black stones stands on the right side of a calm reflective surface, with a single round stone placed beside it on the left. The sky glows with warm pastel colors at sunset, and the water reflects the tranquil scene.

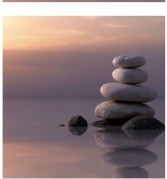

Target Prompt: A small stack of smooth [white] stones stands on the right side of a calm reflective surface, with a single round stone placed beside it on the left. The sky glows with warm pastel colors at sunset, and the water reflects the tranquil scene.

Figure 13: **Failure Case**

### D.2    EXTENDING TO VIDEO EDITING

Video generation (Ma et al., 2024a; 2026; 2025f;e;b; 2024b; 2022; Chen et al., 2025d; Wu et al., 2024; Li et al., 2025; Gao et al., 2025; Song, 2022; Song et al., 2023; Qiu et al., 2024; 2025c; Zhang et al., 2025f; 2024a; Song et al., 2025; Qiu et al., 2025b;d;a; Song & Zhang, 2022; Zhang et al., 2025e;c;b;a;d; Fan et al., 2025; Cai et al., 2025) is an important research direction for practical application. We also explore extending our shape-aware editing framework to the video domain using Wan 2.1 (Wan et al., 2025), an open-source video generation model that uses Rectified Flow. While our method can in principle be applied to all frames, we find that the temporal dimension introduces a major challenge, where the TDM becomes much less stable and effective when extended across time, as shown in Figure 14. In particular, the spatial editing regions indicated by TDM often fluctuate across frames, leading to inconsistent or incomplete transformations in the resulting video. Since a well-defined and temporally consistent TDM is crucial for successful editing, future work may consider strategies such as temporally-aware TDM construction, or explicit disentanglement of spatial and temporal components in the denoising trajectory.

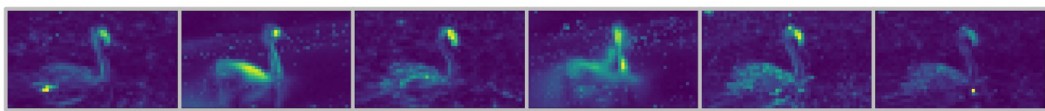

Figure 14: **TDM of Wan2.1 video editing at a single timestep across different frames**.

### D.3    EXTENDING TO AGENT-BASED EDITING

An important future direction is to extend our framework toward an agent-based video editing paradigm (Lin et al., 2025a; Wang et al., 2025c;b;a; Lin et al., 2025b;c; Liu et al., 2025b; Zhao et al., 2024; 2026; 2025; Chen et al., 2025b;a). Instead of treating editing as a single-pass generation problem, we envision a multi-agent system in which specialised agents are responsible for temporal consistency verification, semantic alignment, motion-aware refinement, and user-intent interpretation. Through iterative reasoning and tool invocation (e.g., tracking, depth estimation, or diffusion-based refinement modules), such agents could decompose complex editing instructions into structured sub-tasks and dynamically adjust intermediate results. This formulation would enable adaptive long-horizon editing, reduce error accumulation across frames, and improve controllability under ambiguous prompts. Moreover, integrating memory mechanisms for cross-frame state tracking may further enhance temporal coherence and enable interactive, feedback-driven video manipulation.

## E    MORE EDITING RESULTS

We present additional shape-aware editing results in Figure 15 and Figure 16. We also present general task editing results in Figure 17.

## F    THE USAGE OF LARGE LANGUAGE MODELS

In this paper, the usage of the LLM mainly falls into the following aspects:

- **Grammar checking and format optimization**: In the paragraphs of the paper, LLMs are used for grammar error checking and format checking of charts and figures.
- **Language polishing**: The text description part of the paper uses LLMs to polish and optimize the language expression.
- **Prompt Generation**: We use Qwen-2.5-VL to generate paired source and edit prompts for images in our constructed benchmark."
- All authors are responsible for the content generated by the LLMs.

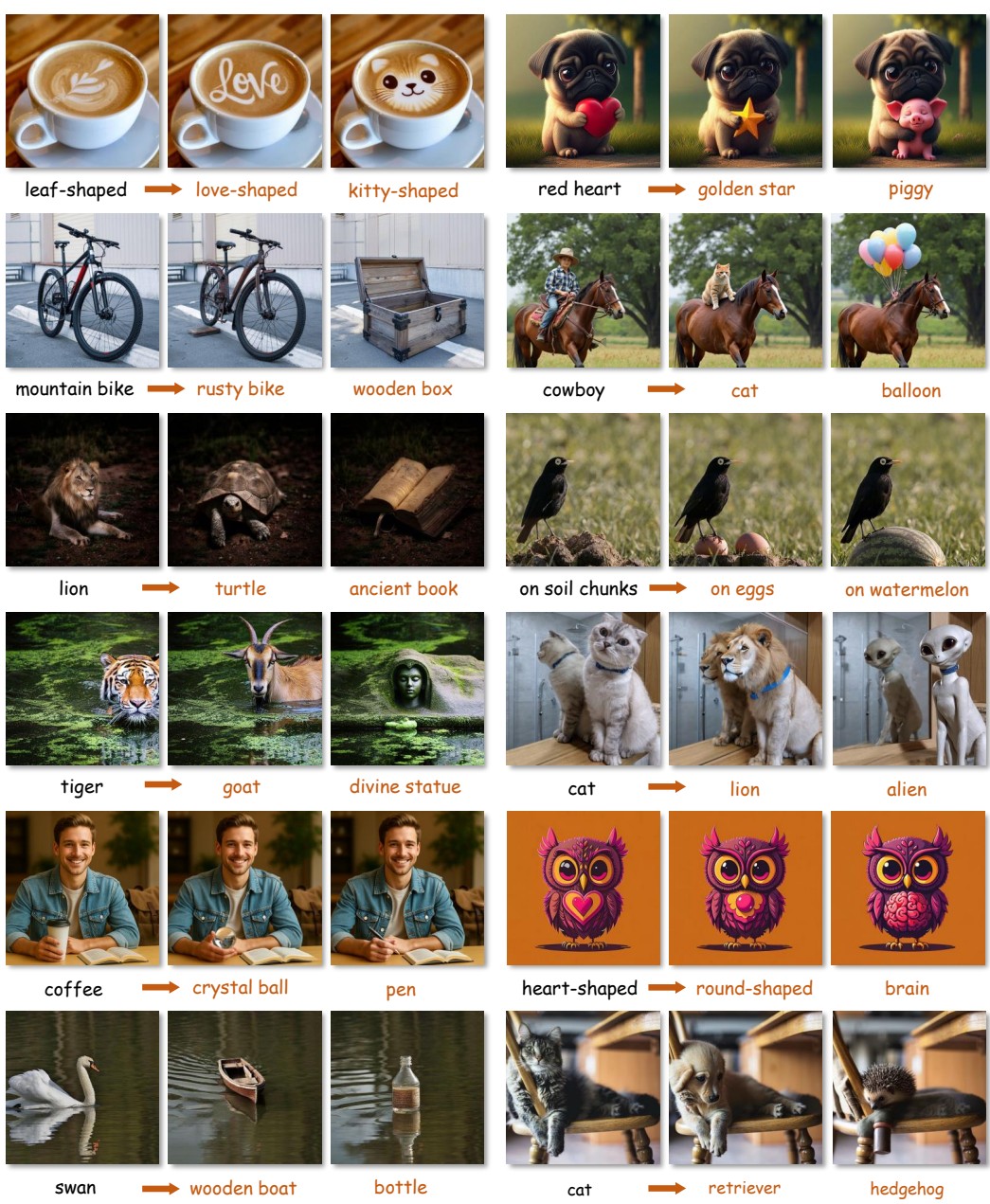

Figure 15: **Additional Editing Results**

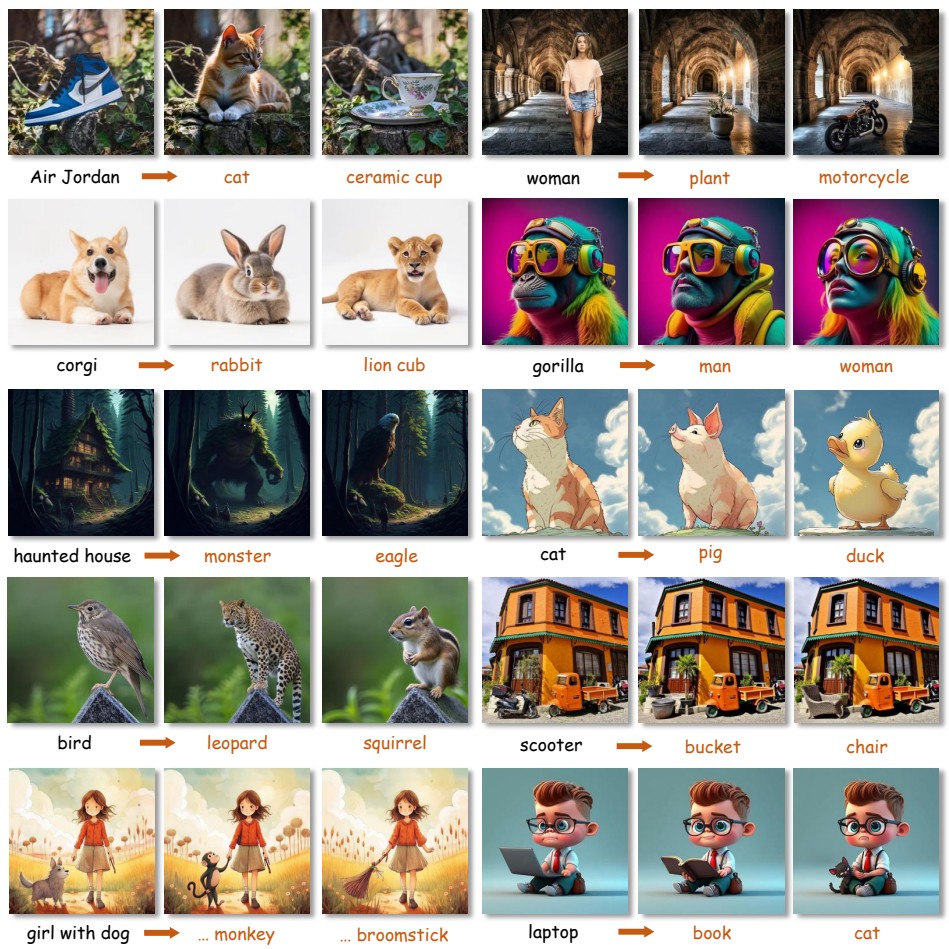

Figure 16: **Additional Editing Results**

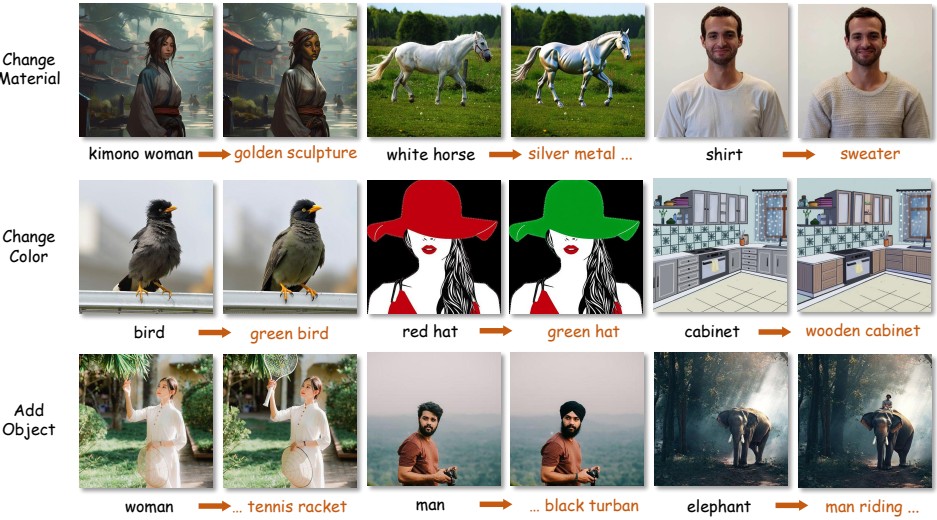

Figure 17: **Additional Editing Results**

Table 5: **Example 1: Image–prompt pairs in** *ReShapeBench*

| Image | Prompts |
|---|---|
| 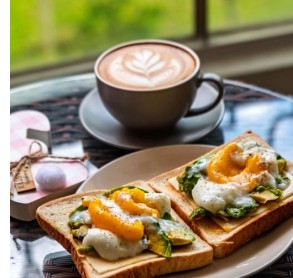 | **Source Prompt:** A single Air Jordan 1 high-top in University Blue rests elegantly atop a weathered tree stump, its bold panels and crisp lines catching the golden afternoon light. The rich blue leather, white toe box, black laces, and dark grey Swoosh stand out sharply against the rough bark beneath, every stitch and texture rendered with photorealistic clarity. Behind it, a soft blur of ivy leaves and tangled branches melts into warm bokeh, hinting at a quiet forest edge bathed in dappled sunlight. The entire scene feels effortlessly stylish — where streetwear meets nature, serene yet striking, captured with cinematic depth and organic warmth. |
| | **Edit Prompt 1:** A sleek tabby cat rests elegantly atop a weathered tree stump, its graceful curves and soft fur catching the golden afternoon light. The rich orange-brown coat, white chest patch, black-tipped ears, and whiskers stand out sharply against the rough bark beneath, every strand rendered with photorealistic clarity. Behind it, a soft blur of ivy leaves and tangled branches melts into warm bokeh, hinting at a quiet forest edge bathed in dappled sunlight. The entire scene feels gently alive — where wild nature cradles quiet companionship, serene and soulful, captured with cinematic depth and organic warmth. |
| | **Edit Prompt 2:** A delicate porcelain teacup with matching saucer rests gracefully atop a weathered tree stump, its fine floral patterns and golden rim catching the golden afternoon light. The white ceramic surface, intricate pink blossoms, green leaves, and gilded edges stand out sharply against the rough bark beneath, every detail rendered with photorealistic clarity. Behind it, a soft blur of ivy leaves and tangled branches melts into warm bokeh, hinting at a quiet garden edge bathed in dappled sunlight. The entire scene feels timelessly elegant — where refined craftsmanship meets rustic nature, serene yet sophisticated, captured with cinematic depth and organic warmth. |

Table 6: **Example 2: Image–prompt pairs in** *ReShapeBench*

| Image | Prompts |
|---|---|
| | **Source Prompt:** A steaming cup of latte with delicate leaf art sits beside two slices of avocado toast topped with runny yolks and fresh herbs, all bathed in soft morning light from a nearby window. The creamy foam, golden crust, vibrant green leaves, and glistening egg yolk stand out sharply against the rustic ceramic plate, every crumb and droplet rendered with photorealistic clarity. Behind it, a gentle blur of lush green foliage outside the glass pane melts into warm bokeh, hinting at a quiet garden waking under diffused daylight. The entire scene feels gently inviting — where simple pleasures meet slow mornings, serene yet richly textured, captured with cinematic depth and organic warmth. |
| | **Edit Prompt 1:** A delicate sprig of white wildflowers rests gently beside two slices of avocado toast topped with runny yolks and fresh herbs, all bathed in soft morning light from a nearby window. The fragile petals, dewy leaves, golden crust, and glistening egg yolk stand out sharply against the rustic ceramic plate, every texture rendered with photorealistic clarity. Behind it, a gentle blur of lush green foliage outside the glass pane melts into warm bokeh, hinting at a quiet garden waking under diffused daylight. The entire scene feels tenderly alive — where nature's grace meets simple nourishment, serene yet richly textured, captured with cinematic depth and organic warmth. |
| | **Edit Prompt 2:** A ripe yellow banana rests casually beside two slices of avocado toast topped with runny yolks and fresh herbs, all bathed in soft morning light from a nearby window. The smooth peel, gentle curve, golden hue, and glistening egg yolk stand out sharply against the rustic ceramic plate, every texture rendered with photorealistic clarity. Behind it, a gentle blur of lush green foliage outside the glass pane melts into warm bokeh, hinting at a quiet garden waking under diffused daylight. The entire scene feels playfully alive — where everyday fruit meets wholesome nourishment, serene yet subtly whimsical, captured with cinematic depth and organic warmth. |

Table 7: **Example 3: Image–prompt pairs in *ReShapeBench***

| Image | Prompts |
|-------|---------|
| 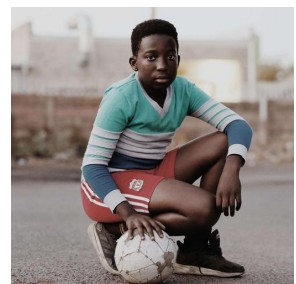 | **Source Prompt:** A young boy crouches low on a quiet street, one hand resting gently on a worn soccer ball, his gaze steady and thoughtful under the soft glow of late afternoon light. 
 His teal striped sweater, red shorts with white stripes, scuffed sneakers, and the cracked leather surface of the ball stand out sharply against the rough asphalt beneath, every thread, crease, and stitch rendered with photorealistic clarity. 
 Behind him, a gentle blur of weathered brick walls and distant buildings melts into warm bokeh, hinting at a humble neighborhood bathed in golden-hour haze. 
 The entire scene feels quietly powerful — where childhood dreams meet everyday resilience, serene yet deeply human, captured with cinematic depth and organic warmth. |
| | **Edit Prompt 1:** A young boy crouches low on a quiet street, one hand resting gently on a worn leather backpack beside his knee, his gaze steady and thoughtful under the soft glow of late afternoon light. 
 His teal striped sweater, red shorts with white stripes, scuffed sneakers, and the frayed straps and faded stitching of the backpack all stand out sharply against the rough asphalt beneath, every texture rendered with photorealistic clarity. 
 Behind him, a gentle blur of weathered brick walls and distant buildings melts into warm bokeh, hinting at a humble neighborhood bathed in golden-hour haze. 
 The entire scene feels quietly nostalgic — where school days meet quiet contemplation, serene yet deeply human, captured with cinematic depth and organic warmth. |
| | **Edit Prompt 2:** A young boy crouches low on a quiet street, one hand resting gently on the back of a sleepy dog curled beside his sneaker, his gaze steady and thoughtful under the soft glow of late afternoon light. 
 His teal striped sweater, red shorts with white stripes, scuffed sneakers, and the soft fur, floppy ears, and relaxed posture of the dog all stand out sharply against the rough asphalt beneath, every texture rendered with photorealistic clarity. 
 Behind him, a gentle blur of weathered brick walls and distant buildings melts into warm bokeh, hinting at a humble neighborhood bathed in golden-hour haze. 
 The entire scene feels tenderly alive — where childhood quietness meets loyal companionship, serene yet deeply human, captured with cinematic depth and organic warmth. |

