# OpenReview forum: "Follow-Your-Shape: Shape-Aware Image Editing via Trajectory-Guided Region Control"
_ICLR.cc/2026/Conference — ICLR 2026 Poster_

### Official Review · Reviewer_XzBW · 2025-10-31

**Soundness:** 3
**Presentation:** 3
**Contribution:** 3
**Rating:** 6
**Confidence:** 4

**Summary:**

This paper introduces EditAnyShape, a training- and mask-free image editing framework designed for precise, shape-aware object editing while strictly preserving background content. Existing editing methods struggle with large-scale shape modifications—they either fail to achieve intended shape changes or degrade undesired regions. The authors propose a novel Trajectory Divergence Map (TDM), which localizes editable regions by analyzing token-wise velocity differences between model inversion and prompt-driven editing paths in latent space. This map guides a scheduled Key-Value (KV) injection mechanism, stabilizing initial trajectories, enabling region-selective edits, and maintaining non-target regions. A new benchmark, ReShapeBench, is introduced, containing 120 curated images and paired prompts to evaluate shape-aware editing models. Experiments show that EditAnyShape achieves state-of-the-art performance for both editability and visual fidelity, especially in challenging large-scale shape transformation tasks.

**Strengths:**

- This paper addresses the challenge of large-scale shape transformations in image editing, which is a well-defined problem
- EditAnyShape is a training and mask free that automatically localizes editable regions (using TDM), also new benchmark ReShapeBench is introduced improving the evaluation of shape-aware editing
- This framework achieves state-of-the-art quantitative performance across multiple metrics

**Weaknesses:**

While EditAnyShape presents a well-structured approach to large-scale shape transformation, several aspects remain underexplored or insufficiently analyzed.

- **Lack of analysis on TDM interpretability and reliability.**

    The paper claims that the Trajectory Divergence Map (TDM) can precisely localize editable regions, unlike the noisy cross-attention maps. However, it remains unclear how effectively TDM identifies the user-intended foreground. Although the authors argue that cross-attention–based attention injection is noisy and inconsistent, they do not provide sufficient experimental evidence demonstrating that TDM is indeed cleaner or more reliable.

    It would strengthen the paper to include visual or quantitative comparisons between TDM and cross-attention maps (e.g., attention heatmaps, spatial correlation, or IoU-based analysis). This would clarify whether TDM truly captures semantic differences rather than inheriting similar noise patterns. Conceptually, since TDM is derived from latent velocity differences between source and edit trajectories, it is not immediately intuitive why it would be less noisy or more foreground-focused than traditional cross-attention maps. More empirical validation is needed to justify this claim.

- **Potential limitations of KV injection and region blending.**

    In Section 3.2.2, the framework performs Key-Value (KV) injection across stages 1–3, blending features from inversion and editing trajectories. However, this mechanism may face challenges in cases involving significant structural layout changes or large scale variations (e.g., when a large object is transformed into a much smaller one) Because KV injection reuses structural information from the source trajectory, there is a risk that unintended layout features from the original image could be reintroduced into the edited regions, potentially interfering with the desired transformation.

    A detailed ablation or visualization showing how KV injection affects spatial consistency and layout preservation in such extreme transformation cases would improve the paper’s rigor. It would also be helpful to discuss whether TDM-based modulation sufficiently suppresses unintended KV influence in non-foreground areas.

**Questions:**

- As mentioned in the Weaknesses, TDM is computed as a latent-level difference between the inversion and editing trajectories. However, it is not intuitively clear how such a latent-space divergence can yield a spatially clear and human-interpretable distinction of editable regions. Could the authors provide further analysis or visualization to support this claim? For instance, how does TDM behave compared to traditional cross-attention maps in terms of localization clarity or noise robustness?
- Regarding the Key-Value (KV) injection process: in cases involving significant structural layout changes (e.g., when a large object is transformed into a smaller one or when the object’s spatial footprint drastically shifts), how does the model prevent source layout information from unintentionally being injected into non-target areas? An ablation or visual example illustrating this effect would help clarify the robustness of the proposed injection mechanism.

---

> ### Author Response · Authors · 2025-11-23
>
> Dear Reviewer XzBW,
>
> Thank you for the detailed and insightful review, and we appreciate your careful evaluation of our method's strengths, and we are grateful for the pointed questions regarding TDM interpretability and the limitations of KV injection. Below we address each concern in order. (W = Weakness, Q = Question)
>
> >1. **(W1, Q1) TDM interpretability and reliability:**
>
> In the Appendix C.4 of the revised manuscript, we added a post-hoc comparison between TDM and alternative region-selection signals (cross-attention masks) under the same staged editing strategy. We've also compared the TDM-guided editing mask with the downsampled union of the ground-truth foreground regions (after editing) to validate its reliability. The results show that: (1) the TDM-guided editing mask closely resembles the downsampled union of the ground-truth foreground regions, indicating that TDM provides a reliable and semantically aligned editing signal; and (2) TDM is significantly cleaner and less noisy than cross-attention maps, while also being easier to extract in practice. These findings support TDM as an effective, interpretable, and robust region-localization cue.
>
> >2. **(W2, Q2) Limitations of KV injection:**
>
> Despite the outstanding performance, our method still has a limitation in prompt ambiguity. As shown in Appendix D.1, when prompts differ minimally—for example, changing only color attributes such as "black stone" to "white stone"—the model may incorrectly identify regions that should remain unchanged as editable areas, leading to unintended modifications in non-target regions. However, TDM is designed to localize where edits should and should not occur, as it captures trajectory divergence under two different prompt conditions. Even when an object undergoes large structural changes (for example, boy-to-backpack/soccer ball, birds-to-dragonflies in Figure 1), TDM highlights the correct semantic region without leaking into unrelated areas, demonstrating its effectiveness in handling substantial shape transformations.
>
> We hope these clarifications and additional analyses address your concerns, and the corresponding revisions in the manuscript are marked in _**blue**_. Please feel free to refer to the other updated sections as well, and let us know if you have any further questions. Thank you!

---

### Official Review · Reviewer_bqGG · 2025-11-01

**Soundness:** 2
**Presentation:** 3
**Contribution:** 2
**Rating:** 4
**Confidence:** 4

**Summary:**

This paper proposes **EditAnyShape**, a training-free and mask-free method for shape-aware image editing using flow-based generative models. The method aims to enable large-scale shape manipulation while maintaining background fidelity. The main technical contribution is the Trajectory Divergence Map (TDM), a token-wise spatial map computed from the divergence between the velocity fields of a source inversion trajectory and its edited counterpart. TDM highlights regions of greatest semantic deviation, effectively predicting “where to edit” without an external mask or cross-attention supervision. Editing is then achieved through a Scheduled KV Injection mechanism in three stages: 1) Early-stage unconditional KV reuse for stability; 2) Middle-stage TDM aggregation; and 3) Late-stage localized KV injection guided by the final TDM (optionally assisted by a ControlNet).

In the experimental results, the authors introduce ReShapeBench, a curated dataset of 120 images with 290 prompt pairs emphasizing geometric and structural changes rather than texture edits.

**Strengths:**

1. The paper identifies a significant and challenging problem. Large-scale shape editing is a major failure case for most SOTA methods.
2. Using divergence between source and target flow trajectories to infer editable regions is original and well-motivated. It moves beyond cross-attention saliency or explicit user masks toward a model-intrinsic notion of semantic locality.
3. The proposed approach can be applied to existing pre-trained flow models without finetuning or additional training data. It integrates smoothly into existing editors, e.g., FLUX and KV-Edit.
4. The mathematical definition of TDM in Equations (1-- 4) is consistent, and the scheduled KV injection is clearly described and empirically ablated.
5. The provided visual examples demonstrate strong control over shape deformation and good background preservation, especially in scenes where previous mask-free editors distort context.
6. The contributed benchmark, ReShapeBench, is expected to fill an evaluation gap by focusing explicitly on shape-aware editing. The dataset and evaluation protocols are carefully designed and could be useful to the community.

**Weaknesses:**

1. The proposed method employs ControlNet guidance (depth/canny maps) during the final editing stage to preserve structure and edges. However, none of the baselines (FlowEdit, RF-Solver, KV-Edit, MasaCtrl, DiT4Edit, etc.) use ControlNet or equivalent structural conditioning. As ControlNet introduces a strong external geometric prior, the resulting improvements in PSNR, LPIPS, and boundary fidelity cannot be attributed solely to the proposed TDM mechanism. As a result, the unfair bassline comparison could invalidate the main quantitative claims in Table 1.
2. The technical novelty is incremental. The Trajectory Divergence Map is a new signal, but the rest of the framework (KV injection, flow editing, scheduled feature blending) extends existing works such as KV-Edit and Stable-Flow. Hence, the novelty lies primarily in the source of the region-control signal (velocity divergence), not in the editing pipeline itself.
3. The constructed ReShapeBench dataset, while valuable, is relatively small (120 images). Its representativeness for general photo editing, especially non-synthetic images, remains uncertain.
4. The presentation lacks runtime and memory analysis. In particular, TDM computation requires per-timestep velocity differences and KV caching, likely increasing GPU memory. The paper also does not quantify overhead relative to baselines.
5. As the proposed method is a training-free, language-driven system, it would be helpful to show cases where TDM misidentifies regions or when prompt wording yields ambiguous edits.

**Questions:**

The authors are suggested to respond to those raised in **Weaknesses.**

---

> ### Author Response · Authors · 2025-11-23
>
> Dear Reviewer bqGG,
>
> Thank you for the careful and comprehensive evaluation. We are glad that the significance of shape-aware editing and the conceptual grounding of TDM were acknowledged, and we respond to each of the raised weaknesses in detail below. (W = Weakness, Q = Question)
>
> >1. **(W1) Fairness concern regarding ControlNet:**
>
> Thank you for raising the fairness concern. We now report an additional experiment in Table 1 and below where ControlNet branches are disabled for every method, including ours. Under this setting, EditAnyShape still surpasses baseline methods on PSNR, LPIPS, and CLIP similarity. This confirms that the performance improvement stems from TDM-guided localization. In our full pipeline, ControlNet provides additional flexibility and more precise structural control during large shape modifications, functioning as an enhancement strategy that improves the controllability and stability of the final edits. This makes TDM still the core contribution and the primary source of performance gain. You can refer to Figure 15, 16, and 17 for more visualizations.
>
> **Table 1: Quantitative comparison with state-of-the-art methods on ReShapeBench and PIE-Bench. (add the quantitative comparison with FLUX.1 Fill, and our method without ControlNet)**
>
> | Methods               |   ReShapeBench    |                             |             |                |     PIE-Bench     |                             |             |                |
> | :-------------------- | :---------------: | :-------------------------: | :---------: | :------------: | :---------------: | :-------------------------: | :---------: | :------------: |
> |                       | **Image Quality** | **Background Preservation** |             | **Text Align** | **Image Quality** | **Background Preservation** |             | **Text Align** |
> |                       |       AS ↑        |           PSNR ↑            | LPIPS×10³ ↓ |   CLIP Sim ↑   |       AS ↑        |           PSNR ↑            | LPIPS×10³ ↓ |   CLIP Sim ↑   |
> | **Diffusion-based**   |                   |                             |             |                |                   |                             |             |                |
> | MasaCtrl              |       5.83        |            23.54            |   125.36    |     20.84      |       5.61        |            21.58            |   130.71    |     19.53      |
> | PnPInversion          |       6.11        |            24.77            |   102.91    |     19.23      |       5.94        |            22.69            |   108.43    |     24.62      |
> | Dit4Edit              |       6.14        |            24.36            |    83.75    |     22.66      |       6.03        |            22.74            |    97.65    |     23.87      |
> | **Flow-based**        |                   |                             |             |                |                   |                             |             |                |
> | RF-Edit               |       6.52        |            33.28            |    17.53    |     30.41      |       6.49        |            31.97            |    15.34    |     29.67      |
> | FlowEdit              |       6.42        |            32.46            |    18.92    |     28.94      |       6.37        |            32.68            |    16.42    |     28.93      |
> | KV-Edit               |       6.51        |            34.73            |    16.42    |     26.97      |       6.47        |            33.45            |    13.72    |     28.14      |
> | FLUX.1Fill            |       6.32        |            31.57            |    19.04    |     28.75      |       6.33        |            32.76            |    17.43    |     26.59      |
> | FLUX.1Kontext         |       6.53        |            32.91            |    18.35    |     28.53      |       6.47        |            34.91            |    14.62    |     28.79      |
> | **Ours**              |                   |                             |             |                |                   |                             |             |                |
> | Ours (w/o ControlNet) |       6.52        |            34.85            |    9.04     |     32.97      |       6.49        |            35.62            |    9.74     |     32.47      |
> | **Ours (Full Model)** |     **6.57**      |          **35.79**          |  **8.23**   |   **33.71**    |     **6.55**      |          **36.02**          |  **8.34**   |    **33.5**    |

---

> ### Author Response · Authors · 2025-11-23
>
> >2. **(W2) Regarding novelty:**
>
> We acknowledge that KV injection and scheduled feature blending have been seen in prior works. However, our contribution is not a minor extension but the introduction of a new region-localization signal derived from rectified-flow theory. TDM is **the first deterministic, model-intrinsic signal** that identifies where the velocity field of an RF model predicts semantic deviation.  This directly addresses the core limitation of existing flow-based editors: without a reliable region-localization signal, methods like KV-Edit and Stable-Flow can only operate globally or rely on pre-defined masks, limiting their applicability in flexible and truly shape-aware editing.
>
> TDM solves this problem by being (i) training-free, (ii) mask-free, (iii) fully prompt-guided, derived directly from the rectified-flow velocity field itself. This allows KV injection and feature blending to become spatially selective, which is essential for enabling mask-free shape editing.  In this sense, while we build upon KV-based editing mechanisms, the novelty lies in the new capability derived from TDM: precise, mask-free shape editing that was not achievable with existing signals.
>
> >3. **(W3) ReShapeBench dataset size and representativeness:**
>
> Although ReShapeBench contains 120 base images, each image is paired with two shape transformations, yielding 290 evaluation cases that cover nature, animals, indoor scenes, outdoor scenes, and multi-object cases. We collected the data from high-resolution photographs rather than synthetic renders, and the prompts follow a four-sentence template to ensure consistent difficulty. The dataset is meant to complement (for this task), not replace, broader suites such as PIE-Bench. To address the concern about representativeness, we now include PIE-Bench results (Table 1 and above), which span significantly more categories and editing types. EditAnyShape consistently outperforms baselines on this large-scale benchmark, demonstrating that the method generalizes well beyond our own dataset.
>
> >4. **(W4) Runtime and memory analysis:**
>
> We clarify that the TDM computation and KV caching do not increase GPU memory usage, as both the KV features and TDM maps are stored in CPU memory. On an NVIDIA A100 (40 GB), the runtime for a 28-step editing process is approximately 65.3 seconds. The CPU memory consumption for storing KV features and TDM maps is around 12 GB, while the GPU memory usage remains stable at approximately 25 GB, comparable to standard flow-based editing pipelines. These measurements quantify the overhead introduced by TDM, and the details are provided in Appendix C.3.
>
> >5. **(W5) Failure cases:**
>
> Thanks for pointing out this issue. Yes, as our method is purely prompt-guided, it is naturally influenced by prompt ambiguity. We now provide explicit failure cases in Appendix D.1. When prompts have low distinguishing power or are not specific enough, for example, changing a color attribute from black to white, TDM may misidentify regions that should remain unchanged, leading to unintended edits in non-target areas. These cases illustrate the sensitivity of our system to subtle prompt variations.
>
> We hope these clarifications and additional analyses address your concerns, and the corresponding revisions in the manuscript are marked in _**blue**_. Please feel free to refer to the other updated sections as well, and let us know if you have any further questions. Thank you!

---

> ### Author Response · Authors · 2025-11-27
>
> Dear reviewer bqGG,
>
> Thanks for your valuable feedback on our paper.
>
> We wanted to follow up regarding the rebuttal we posted earlier addressing all your comments (including the points on ControlNet fairness, novelty, dataset, runtime/memory, and failure cases). If there are any remaining concerns or if anything in our clarifications requires further explanation, please feel free to let us know. We would be glad to provide any additional details you may need.
>
> Best,
>
> Authors

---

### Official Review · Reviewer_KcbZ · 2025-11-04

**Soundness:** 3
**Presentation:** 4
**Contribution:** 3
**Rating:** 6
**Confidence:** 4

**Summary:**

The paper introduces EditAnyShape, a training-free, mask-free framework for shape-aware image editing built on the FLUX.1 rectified-flow model. It targets large geometric and structural changes of foreground objects driven by text prompts, while maintaining high fidelity in non-edited regions. The core idea is a Trajectory Divergence Map (TDM) that measures token-wise velocity differences between inversion and editing trajectories to localize where semantics should change, which then guides staged KV injection and ControlNet conditioning. The authors also propose ReShapeBench, a curated benchmark of 120 images and 290 shape-editing cases, and show that their method outperforms diffusion-based and flow-based baselines on aesthetic quality, text alignment, and background preservation.

**Strengths:**

- Trajectory-based region control. TDM offers a conceptually grounded way to localize editable regions by exploiting velocity differences in rectified-flow trajectories, avoiding reliance on noisy attention maps or external segmentation.

- Training-free and mask-free pipeline. The method operates purely with a pre-trained FLUX model and does not require hand-drawn or model-generated masks, which reduces annotation overhead and simplifies deployment.


- Consistent empirical gains. Across ReShapeBench, the approach improves aesthetic scores, maintains better background similarity (PSNR/LPIPS), and achieves stronger text-image alignment than diffusion- and flow-based baselines.


- Targeted ablations. Parameter studies on early stabilization length and ControlNet timing clarify how different components contribute and highlight reasonable default settings.


- Task-focused benchmark. ReShapeBench explicitly emphasizes shape and structural transitions with clear criteria and curated prompts, helping the community study this specific editing regime.

**Weaknesses:**

- Global PSNR and LPIPS cannot disentangle background preservation from foreground edits, so the core claim of “preserving non-edited regions” is only indirectly tested. Some region-restricted metrics would provide stronger evidence.

- The paper does not empirically compare TDM to simpler region-selection strategies (e.g. DiffEdit-style prediction differences, cross-attention masks) when used within the same staged KV injection scheme, leaving the unique benefit of TDM somewhat under-quantified.

- Runtime and memory costs introduced by TDM accumulation and extra passes are not summarized in the main text.

- The paper primarily shows successful edits and does not systematically discuss where TDM fails (e.g. thin structures, cluttered scenes, overlapping objects), which would be valuable for practitioners.

**Questions:**

- Could you report background-only and foreground-only PSNR/LPIPS (e.g. using TDM masks or manual masks on a subset) to more directly validate background preservation?

- How does TDM compare to a DiffEdit-style prediction-difference mask or to cross-attention-based masks when combined with your staged KV injection and ControlNet schedule?

---

> ### Author Response · Authors · 2025-11-23
>
> Dear Reviewer KcbZ,
>
> Thank you for the careful evaluation and for highlighting both the strengths of our trajectory-guided editing framework and the points that require further analysis. Below we provide responses to each comment and clarify the points raised. (W = Weakness, Q = Question)
>
> >1. **(W1, Q1) PSNR/LPIPS for background preservation:**
>
> For shape-editing tasks, pixel-accurate foreground masks are inherently unavailable because the foreground object undergoes structural deformation, and no ground-truth target exists to define the edited region precisely. Therefore, following common practice in editing evaluation, region-restricted metrics are typically computed only on the background, where the notion of "unchanged region" is well-defined. To approximate the background region under our task categorization (see Appendix B.1), which assumes that the transformed object retains its spatial anchor and contextual role, we follow standard heuristics and mask out the foreground using task-specific bounding boxes (see Appendix B.1 and B.3). This allows us to compute background-only PSNR/LPIPS, which directly reflects preservation quality in non-edited regions. In short, foreground-only metrics are not meaningful in this setting because the edited foreground lacks ground-truth correspondence.
>
> >2. **(W2, Q2) Comparison against alternative region-selection strategies:**
>
> We have added a post-hoc comparison between TDM and cross-attention masks under the same staged editing strategy. We have also compared the TDM-guided editing mask with the downsampled union of the ground-truth foreground regions (after editing). The results show two advantages of TDM: (1) the TDM-guided editing mask closely resembles the downsampled union of the ground-truth foreground regions, indicating that TDM provides a reliable and semantically aligned editing signal; and (2) TDM is significantly cleaner and less noisy than cross-attention maps, while also being easier to extract in practice. These findings support TDM as an effective, interpretable, and robust region-localization cue.
>
> Regarding the comparison with DiffEdit, we note that the mask-generation mechanism in DiffEdit has several methodological limitations that motivate the need for a more principled alternative. First, DiffEdit constructs its mask by averaging prediction differences across multiple timesteps and applying a global threshold of 0.5. This aggregation strategy is not compatible with the operational dynamics of diffusion/flow models, and it overlooks the fact that a token that remains stable at one timestep may still undergo changes at later stages. In their case, meaningful structural variations can be diluted, making the resulting mask less aligned with the intended editing behavior. In contrast, our staged TDM-guided formulation explicitly tracks trajectory evolution and provides a more rational and interpretable localization signal.
>
> Second, DiffEdit performs direct latent blending using the binary mask, which may introduce boundary discontinuities or unnatural transitions because hard latent substitution does not respect the model's attention structure. Our method instead regulates region-specific behavior through KV feature reuse, allowing the model to maintain contextual coherence while applying edits. As a result, region transitions are smoother and more semantically consistent.
>
> >3. **(W3) Runtime and Memory:**
>
> We clarify that KV features and TDM maps are stored in CPU memory rather than GPU memory, which may be misunderstood by the reviewer. On an NVIDIA A100 (40 GB), the running time for 28 NFEs is approximately 65.3 seconds. The CPU memory usage for storing KV features and TDM maps is approximately 12 GB, while the GPU memory usage remains stable at around 25 GB throughout the editing process. Details are provided in Appendix C.3.
>
> >4. **(W4) Failure Cases:**
>
> We have updated the failure cases in Appendix D.1. In short, ambiguous or weakly specified prompts can cause the model to misinterpret the intended editing target or apply insufficient semantic change, resulting in diffuse or inconsistent outputs. These cases illustrate that the framework relies on clear, discriminative textual instructions to reliably guide trajectory divergence and achieve precise region-localized edits.
>
> We appreciate the reviewer's thoughtful comments and hope that the additional analyses, comparisons, and clarifications in the revised manuscript (marked in _**blue**_) adequately address all concerns. Please feel free to refer to the other updated sections as well, and let us know if you have any further questions. Thank you!

---

### Official Review · Reviewer_b8cf · 2025-11-09

**Soundness:** 3
**Presentation:** 3
**Contribution:** 4
**Rating:** 6
**Confidence:** 4

**Summary:**

This paper focuses on improving precision of large-scale shape editing in images. In contrast to binary segmentation masks or cross-attention masks, which tend to be noisy and unreliable for shape transformation, TDM is based on the semantic difference between source and target denoising trajectories.

The core contributions are:
(1) EditAnyShape -- a novel training-free and mask-free approach for controllable object shape editing in images.
(2) A selective KV injection mechanism based on a Trajectory Divergence Map (TDM). This TDM serves as a guide for more precise and localized KV injection, preventing unintended edits.
(3) A Scheduled KV Injection approach to only introduce TDM-guided editing in later stages of the denoising process, in order to prevent instability in the early, high-noise denoising stages.
(4) A new benchmark, ReShapeBench, consisting of 120 (image, prompt) pairs for shape-aware editing.

**Strengths:**

- The general insight of the paper to use magnitude of trajectory difference for localizing edits is both clever and intuitive.
- The methods were generally well-motivated and clearly explained.
- The paper is overall well-written and polished.

**Weaknesses:**

- Figure 2 is quite confusing. For example, is that row in the top right a legend for left and right sides of the figure? If so, it could be labelled more clearly. Also, for the bottom figure, the outline colors of the frames (particularly blue and orange) are very hard to notice.
- The quantitative evaluation only includes results on the paper's custom evaluation dataset, but does not report results of a third-party editing dataset, such as PIE-Bench. Although PIE-Bench does not isolate shape changes, it would be more convincing to see if this method can perform on-par with other editing methods on an unbiased outside dataset.
- In some qualitative comparisons of Figure 6, it looks like other methods seem similar to the EditAnyShape framework. For example, in the third-to-last row, KV-Edit looks like it is making the figure 8 ball and preserving the background well. What is the inherent benefit over kv-edit (also training-free)?
- The ablation study doesn't actually compare to the base Flux.1-dev model, without adding EditAnyShape framework. It would be informative to know how well does the base model do for editing by default?

**Questions:**

- In Figure 2 (left), what is "Other shape editing"?
- Is there some way to visualize the TDM and compare it to the ground truth difference in foreground masks of the source and target images?
- Perhaps Section 4.2 about benchmark construction would better belong in the Methods section or its own section, rather than in Experiments?
- Section 4.2: What is the purpose of the third "general evaluation set"? Is it that subsets 1 and 2 are for training and subset 3 is for testing?
- Since this approach is training-free, it would be great to see it applied to other open-source flow-matching models, like

---

> ### Author Response · Authors · 2025-11-23
>
> Dear Reviewer b8cf,
>
> Thank you for the thoughtful and constructive feedback and suggestions, and we appreciate the positive assessment of our paper's contribution. Below we address each point in order. (W = Weakness, Q = Question)
>
> >1. **(W1, Q1) Clarification and revision of Figure 2:**
>
> Thank you for pointing out the ambiguity in Figure 2. We have fully redrawn the figure to improve clarity and added it to the revised version. In particular, we removed the "other shape editing" examples and moved those cases to the appendix. We also improved the alignment between the TDM visualizations and the corresponding editing stages. The legend, color scheme, and all related references in the text have been updated accordingly.
>
> >2. **(W2, Q4) PIE-Bench comparison, Using general evaluation sets:**
>
> For clarity, we explain how the evaluation data are organized. We first construct two subsets of images—one for single-object cases and one for multi-object cases. For the purpose of evaluating general editing ability, we select representative cases from these two subsets and combine them with public cases from PIE-Bench to form our general evaluation set. As our method is entirely training-free, this set is used purely for evaluation. However, as you pointed out, a more comprehensive and unbiased evaluation set is needed to cover the diverse range of editing task types. To address this concern, we additionally conducted experiments on PIE-Bench to evaluate its general editing ability. The PIE-Bench results are now included in Table 1 of the revised manuscript and below. We also provide visualizations in Figures 15, 16, and 17. Our method maintains competitive editing performance while still offering shape-aware control, supporting the generality of our approach.
>
> **Table 1: Quantitative comparison with state-of-the-art methods on ReShapeBench and PIE-Bench.**
>
> | Methods | ReShapeBench | | | | PIE-Bench | | | |
> |:--------|:------------:|:-:|:-:|:-:|:--------:|:-:|:-:|:-:|
> | | **Image Quality** | **Background Preservation** | | **Text Align** | **Image Quality** | **Background Preservation** | | **Text Align** |
> | | AS ↑ | PSNR ↑ | LPIPS×10³ ↓ | CLIP Sim ↑ | AS ↑ | PSNR ↑ | LPIPS×10³ ↓ | CLIP Sim ↑ |
> | **Diffusion-based** | | | | | | | | |
> | MasaCtrl | 5.83 | 23.54 | 125.36 | 20.84 | 5.61 | 21.58 | 130.71 | 19.53 |
> | PnPInversion | 6.11 | 24.77 | 102.91 | 19.23 | 5.94 | 22.69 | 108.43 | 24.62 |
> | Dit4Edit | 6.14 | 24.36 | 83.75 | 22.66 | 6.03 | 22.74 | 97.65 | 23.87 |
> | **Flow-based** | | | | | | | | |
> | RF-Edit | 6.52 | 33.28 | 17.53 | 30.41 | 6.49 | 31.97 | 15.34 | 29.67 |
> | FlowEdit | 6.42 | 32.46 | 18.92 | 28.94 | 6.37 | 32.68 | 16.42 | 28.93 |
> | KV-Edit | 6.51 | 34.73 | 16.42 | 26.97 | 6.47 | 33.45 | 13.72 | 28.14 |
> | FLUX.1Fill | 6.32 | 31.57 | 19.04 | 28.75 | 6.33 | 32.76 | 17.43 | 26.59 |
> | FLUX.1Kontext | 6.53 | 32.91 | 18.35 | 28.53 | 6.47 | 34.91 | 14.62 | 28.79 |
> | **Ours** | | | | | | | | |
> | Ours (w/o ControlNet) | 6.52 | 34.85 | 9.04 | 32.97 | 6.49 | 35.62 | 9.74 | 32.47 |
> | **Ours (Full Model)** | **6.57** | **35.79** | **8.23** | **33.71** | **6.55** | **36.02** | **8.34** | **33.51** |
>
> >3. **(W3) Difference from KV-Edit:**
>
> We agree that certain qualitative cases may appear visually similar, but KV-Edit has two fundamental limitations that prevent it from handling large shape transformations:
>
> 1. **Dependence on external masks:** KV-Edit requires manually provided binary masks, which introduces an extra input condition.
>
> 2. **Limited ability to perform large-geometry editing:** For large shape changes, mask-guided feature reuse fails to capture the full extent of geometric deformation needed. As shown in examples such as the parrot and dragon rows, KV-Edit introduces ghosting artifacts when dealing with significant structural changes. Our method, being mask-free and guided purely by trajectory divergence, avoids these issues and preserves background fidelity more effectively.
>
> Thus, the inherent advantage of our approach lies in mask-free localization and better handling of large shape transformations.

---

> ### Author Response · Authors · 2025-11-23
>
> >4. **(W4) Comparison with the base FLUX.1-dev model:**
>
> The FLUX.1-dev model is designed for image generation rather than editing, so it is hard to compare directly. However, we address the reviewer's concern by clarifying that our comparison is fair and appropriate:
>
> 1. **Training-free editing baselines on FLUX.1-dev:** Both RF-Edit and KV-Edit are implemented directly on top of FLUX.1-dev. Therefore, comparing against them effectively reflects the editing behavior that FLUX.1-dev can achieve under training-free editing settings.
>
> 2. **FLUX.1 Kontext as a training-based editing baseline:** FLUX.1 Kontext is a training-based editing-focused variant built on FLUX. Including it provides a stronger editing baseline and further contextualizes the performance of our training-free method.
>
> 3. **FLUX.1 Fill (inpainting):** Since inpainting is the closest built-in editing mechanism available in FLUX.1-dev, we add the comparison of FLUX.1 Fill. This serves as a practical proxy for the native editing capability of the base FLUX model.
>
> These direct and indirect comparisons collectively address the reviewer's concern regarding the base model's editing behavior.
>
> >5. **(Q2) Correlation between TDM and actual edited regions:**
>
> We add a post-hoc comparison in Appendix C.4. Although our method does not use masks during inference, we can still examine whether the TDM correlates with the actual edited regions. We therefore extract foreground/background masks using SAM for both the source and the edited images, downsample them to the token-level resolution, and compare the downsampled masks with the TDM. This visualization shows that the TDM indeed reflects the semantic regions where edits occur. These results are included in Appendix C.4.
>
> >6. **(Q3) Benchmark construction section:**
>
> Thank you for the suggestion. We reorganized the structure and moved the benchmark construction discussion after the Methods section as an independent section for clarity.
>
> >7. **(Q5) Extension to other flow-matching models:**
>
> We agree that exploring other open-source flow-matching models is an interesting future direction. However, our method is particularly well-suited for RF-based models, because the TDM relies on the linear and geometry-preserving properties of the rectified flow trajectory. In RF, the deterministic straight-line flow and its associated velocity field provide a meaningful structure for interpreting divergence and for relating trajectory deviation to semantic editing regions. A direct extension would not yield a fair or meaningful comparison without additional methodological adaptations. We therefore focus the experiments on RF-based models, where our method is theoretically grounded and most applicable.
>
> We appreciate the reviewer's detailed feedback and the opportunity to clarify these points. The revisions are incorporated in the updated manuscript (marked in _**blue**_). We hope these changes address all concerns. Please feel free to refer to the other updated sections as well, and let us know if you have any further questions. Thank you!

---

### Author Response · Authors · 2025-12-02

We would like to express our sincere gratitude to the Reviewers, Area Chairs, and Program Chairs for the time and effort invested in evaluating our submission. We appreciate the detailed comments, constructive suggestions, and thoughtful critiques that guided our revision process. Below, we briefly summarize the feedback received and the adjustments implemented in the updated manuscript.

Our work received positive recognition from the reviewers in several key areas, including:

- **Clear problem definition:** Reviewers agreed that large-scale shape transformation is a well-defined and important problem that existing editing models struggle with (bqGG, XzBW).

- **Effectiveness of TDM:** The trajectory-based region localization signal was noted as intuitive (b8cf), well-motivated (bqGG), and conceptually grounded (KcbZ).

- **Training- and mask-free design:** the method operates directly on pre-trained flow models without additional training or external masks, making it practical and applicable (KcbZ, bqGG, XzBW).

- **Benchmark contribution:** ReShapeBench was recognized as addressing a gap in evaluating shape-aware (task specific) editing (KcbZ, bqGG, XzBW).

To address the reviewers’ concerns regarding TDM interpretability, comparison fairness, dataset representativeness, and the analysis of runtime and failure cases, we have added additional experiments (including ControlNet-disabled evaluations and PIE-Bench results), provided new visual and quantitative analyses of TDM, included more detailed hyperparameters as well as runtime and memory information, and reorganized and clarified sections of the manuscript. The revisions in the manuscript are marked in ***blue***. A summary of these changes follows (in the order they appear in the manuscript):

- In response to reviewer b8cf’s suggestions, we clarified Figure 2 in Section 3.1 and reorganized the benchmark construction into a separate section.
- In response to reviewers b8cf and bqGG, we added more baseline method, ControlNet-disabled comparison, and additional experiments on PIE-Bench in Section 5, ensuring more fair and representative evaluations.
- We added more hyperparameter details in Appendix C.2 to support reproducibility.
- In response to reviewers bqGG and KcbZ, we added runtime analysis and memory usage in Appendix C.3.
- In response to reviewers b8cf, KcbZ, and XzBW, we added a post-hoc analysis of TDM in Appendix C.4 to further confirm its effectiveness.
- In response to reviewers KcbZ, bqGG, and XzBW, we clarified limitations and provided additional failure cases in Appendix D.1.

In particular, regarding reviewer bqGG’s concern on novelty, our core contribution is a deterministic, model-intrinsic region localization signal (TDM) that enables precise, spatially selective, and mask-free control for shape-aware editing—capabilities not achievable with prior KV-based pipelines. We hope this work provides a useful step toward more principled region localization in image generation models and helps advance reliable, mask-free shape transformation in image editing.

We thank the reviewers and chairs again for their time and thoughtful evaluation.

Sincerely,

Paper 6089 Authors

---

### Meta-Review · Area_Chair_7dPo · 2025-12-28

**Summary:**

This submission received **largely positive initial scores**, with three reviewers rating the paper at or above the acceptance threshold (Reviewers b8cf, KcbZ, XzBW: all 6) and one reviewer initially leaning marginal reject (Reviewer bqGG: 4). Overall, reviewers acknowledged that the paper addresses an important and well-defined problem and introduces a novel, model-intrinsic region localization signal, namely the Trajectory Divergence Map (TDM).

Across reviews, there is agreement that TDM is a clever and conceptually grounded idea, enabling training-free and mask-free shape-aware editing that is difficult to achieve with existing cross-attention- or mask-based methods. Reviewers also appreciated the introduction of ReShapeBench, which fills a gap in evaluating shape-centric editing scenarios. While concerns were raised regarding fairness of comparisons (ControlNet usage), incremental novelty relative to prior KV-based editors, runtime/memory overhead, and failure case analysis, the rebuttal comprehensively addressed these points through additional experiments (e.g., ControlNet-disabled comparisons, PIE-Bench results), expanded analyses, and clarifications. Based on the overall review sentiment and the effectiveness of the rebuttal, my judgment as AC is to **Accept**.

**Reviewer Concerns:**

### Concerns effectively addressed by the rebuttal

* **Fairness of comparisons due to ControlNet usage**:
  Reviewer bqGG raised a major concern that ControlNet introduces an external geometric prior not used by baselines, potentially confounding quantitative gains. The authors addressed this by adding ControlNet-disabled comparisons for all methods, showing that EditAnyShape still outperforms baselines, thereby isolating the contribution of TDM-guided localization.

* **Lack of evaluation on third-party benchmarks**:
  Reviewers b8cf and bqGG noted that initial quantitative results focused mainly on the proposed ReShapeBench. The authors added PIE-Bench experiments, demonstrating competitive or superior general editing performance while maintaining shape-aware control.

* **Interpretability and reliability of TDM**:
  Reviewers KcbZ and XzBW questioned whether TDM genuinely localizes editable regions better than cross-attention maps. The rebuttal added post-hoc visual and quantitative analyses, including comparisons to cross-attention masks and downsampled foreground regions, showing that TDM is cleaner and more semantically aligned.

* **Runtime and memory analysis**:
  Reviewer bqGG requested explicit runtime and memory costs. The authors clarified that TDM and KV features are stored in CPU memory, and provided concrete runtime and memory numbers (A100 GPU), addressing practicality concerns.

* **Failure cases and limitations**:
  Multiple reviewers requested clearer discussion of when the method fails. The authors added explicit failure cases and clarified that prompt ambiguity is a key limitation, improving transparency.


### Minor concerns that remain outstanding (non-blocking)

* **Incremental nature of the overall pipeline**:
  While TDM itself is novel, some reviewers noted that the surrounding editing framework (KV injection, staged blending) builds on prior work such as KV-Edit and flow-based editors
  *(Reviewer bqGG)*. This does not diminish the contribution but suggests the novelty lies primarily in the region-control signal rather than the full pipeline.

* **Dataset scale and representativeness**:
  ReShapeBench is relatively small (120 images), and although supplemented with PIE-Bench results, broader validation on more diverse real-world editing scenarios could further strengthen the empirical story
  *(Reviewer bqGG)*.

* **Metric granularity for background preservation**:
  Some reviewers suggested that more fine-grained, region-restricted metrics could better disentangle foreground editing from background preservation
  *(Reviewer KcbZ)*.

**Reviewer Scores:**

* **Reviewer b8cf (initial: 6)** → **Likely unchanged**
  Strongly positive on the intuition and novelty of TDM; presentation and evaluation concerns were addressed.

* **Reviewer KcbZ (initial: 6)** → **Likely unchanged**
  Appreciated the conceptual grounding and consistent gains; remaining questions focused on analysis depth.

* **Reviewer XzBW (initial: 6)** → **Likely unchanged**
  Found the problem important and the method effective; interpretability concerns were addressed via added analysis.

* **Reviewer bqGG (initial: 4)** → **Possibly improved, but no post-rebuttal response recorded (4 → 6)**
  Major concerns on fairness, runtime, and novelty were substantively addressed in the rebuttal, though the reviewer did not participate further in discussion.

---

### Decision · Program_Chairs · 2026-01-26

Accept (Poster)